# From End-to-End to Step-by-Step: learning Composable Navigation Primitives for Vision-Language Navigation

## Abstract

Recent Vision-Language Navigation (VLN) research with Multi-modal Large Language Models (MLLMs) has broadly adopted end-to-end training on long-horizon instruction datasets. However, human navigation mainly relies on the sequential execution of simple primitives guided by immediate observations. Our analysis shows that, although VLN models reported achieving promising results on long-horizon instructions, they struggle with basic navigation primitives (e.g., move, change region). To the best of our knowledge, we are the first to point out this phenomenon. To address this, we propose a primitive-based paradigm that first learns core skills and then composes them into long-horizon behaviors. We design a unified data pipeline to construct Vision-Language-Move-Base (VLMB), the first controllable benchmark centered on the move-to primitive, covering 206 scenes and 873 object instances. Based on VLMB, we develop Move-to-Anything, a model equipped with a memory mechanism that balances historical context with current observations. Experiments demonstrate that existing VLN models achieve only a 43.8% success rate in MP3D; our approach reaches 60.6% in MP3D and 71.4% in HM3D, exhibiting substantially stronger compositional generalization. These results highlight the effectiveness of primitive-based learning for building robust and generalizable navigation agents.

## 1 Introduction

Vision-Language Navigation (VLN) aims to develop autonomous agents that can understand natural language instructions and perform navigation tasks effectively. Achieving efficient and stable performance in VLN is a crucial step toward building intelligent robotic systems. Accordingly, substantial research in this field has adopted modular methods, which decompose the navigation pipeline into separate components such as perception, decision-making, planning, and control (Chen et al., 2023; Zhang et al., 2025b; Ma et al., 2025). However, according to the "weakest link" principle, the overall system performance is often limited by its least competent module, and such modular designs frequently hinder effective collaboration between perception and planning.

With recent progress in Multi-modal Large Language Models (MLLMs), end-to-end approaches for VLN have emerged as a promising alternative (Zheng et al., 2024; Cheng et al., 2025). Representative systems such as NaVid (Zhang et al., 2024a) and StreamVLN (Wei et al., 2025) leverage existing VLN instruction datasets (Anderson et al., 2018; Ku et al., 2020; Krantz et al., 2020) and collect visual data from simulators to train general-purpose MLLMs for VLN tasks. These methods significantly improve the alignment between visual perception and language instructions, thereby facilitating applications in embedded AI and human–computer interaction.

Although the field of VLN has made notable progress, current end-to-end VLN models are still primarily trained using instruction paradigms from early datasets such as R2R (Anderson et al., 2018) and RxR (Ku et al., 2020), as well as datasets repeatedly collected or processed based on these foundations (Tan et al., 2019). These paradigms often include complex instructions like: "*Walk to the right of the couch and through the doors to the tan couch. Pass through the dining room and wait outside.*" Such instruction formats often involve high task complexity and long sequences. Though researchers assume that training MLLM on such datasets is enough to accomplish long-horizon

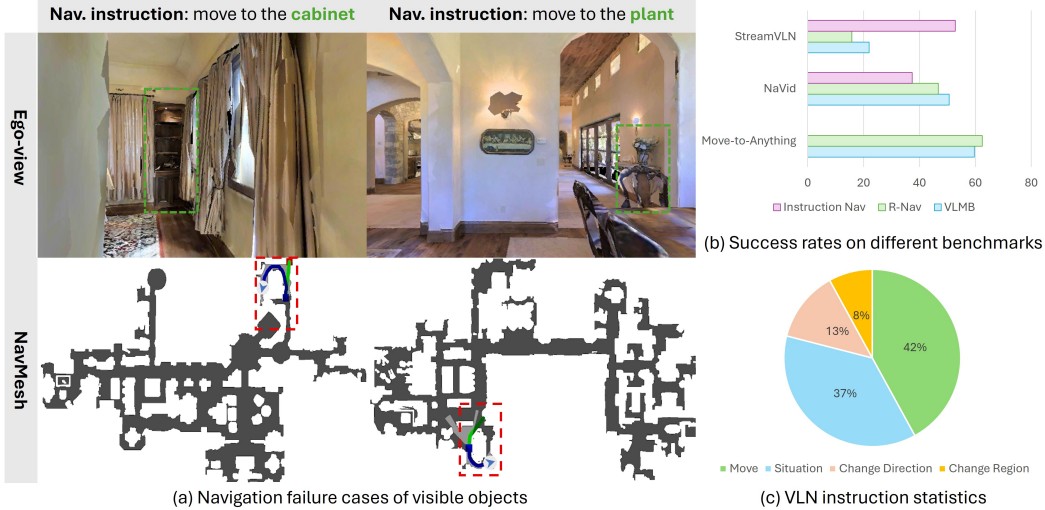

Figure 1: Analysis of existing SOTA models and classic VLN datasets. Figure (a) shows two simple navigation instructions. The NavMesh reveals that the agent stops at incorrect locations. Figure (b) presents the performance of SOTA models on different benchmarks. Figure (c) illustrates the proportional breakdown of instruction types in existing datasets.

navigation tasks, real-world navigation tasks mainly focus on simple, multi-step guidance, such as "*Move to the couch, then move to the tan couch.*", and our experiments show that existing models fail to execute these general but straightforward instructions effectively, as illustrated in Figure 1(a). In reality, while general-purpose MLLMs possess strong visual and linguistic understanding capabilities, they lack foundational navigation skills—the ability to execute correct basic actions even in simple environments. Training them directly on long-horizon datasets causes them to skip this essential foundational training phase. Consequently, the model's output primarily focuses on aligning visual observations with instructions, neglecting core navigation competency. The visualization of several successful case in Appendix B.2 Figure 8 serves as compelling evidence for this viewpoint.

To better understand this issue, we conducted a systematic evaluation of SOTA VLN models. The results reveal that models trained under current paradigms perform poorly on basic single-step navigation tasks. Surprisingly, their success rates on these tasks are even lower than on long-horizon instructions (see Figure 1(b)). We identify two significant limitations in existing datasets: (1) Early datasets were designed for modular VLN methods and emphasize long-horizon instruction following, with limited attention to basic navigation tasks. (2) Existing datasets lack sufficient spatial and semantic coverage, which restricts the model's ability to learn general navigation capabilities.

To overcome these issues, we begin to explore a low-cost, step-by-step training paradigm that enables general-purpose MLLMs to incrementally and reliably acquire fundamental navigation skills.

We performed a statistical analysis of existing instruction sets. We found that, despite their apparent complexity, most instructions can be categorized into four basic navigation primitives: *situation*, *move*, *change direction*, and *change region*. We define these categories as navigation primitives. Among them, *move* is the most common, accounting for 42% of all instructions (see Figure 1(c)). A quantitative examination of the instruction distribution is presented in Appendix B.1 Table 7. At the same time, most navigation tasks can be encapsulated by "move-to" instructions that incorporate spatial and semantic information. For example, the "change region" primitive can be accomplished by combining the "move-to" primitive with spatial relationships between regions to achieve the final navigation task. More interestingly, human navigation reasoning also relies on the sequential execution of single-step primitives based on immediate observations. By moving from one observation point to another, humans achieve long-horizon navigation goals. This motivates our focus on "move-to" training at the primitive-level for MLLMs.

To achieve the step-by-step training paradigm, we introduce Vision-Language-Move-Base (VLMB). We collected an initial set of raw navigation instructions (R-NaV) and extended them to form the

VLMB dataset with semantic and spatial annotations. VLMB automated data generation pipeline integrates 206 semantic scenes from the MP3D (Chang et al., 2017) and HM3D-Semantic (Ramakrishnan et al., 2021) datasets, which comprises over 873 distinct target instances. This dataset is designed to isolate, evaluate, and improve the core "move-to" capability in VLN. The dataset construction begins with a rule-based collection of basic "move-to" instructions. These instructions are then expanded through MLLM-based cross-validation into versions enriched with spatial and semantic information, thereby capturing the diversity and ambiguity of natural language. An interactive verification process ensures high data quality while incorporating spatial and semantic cues to improve the understanding of models of both objects and scenes. Furthermore, we compose navigation primitives to execute basic navigation tasks in a stepwise manner, ultimately achieving controllable long-horizon navigation.

To ensure fair evaluation and validate data quality, we use the same base model (Zhang et al., 2024c) as employed in current SOTA approaches. For the navigation task, we design a hierarchical memory mechanism that combines temporal-semantic embedding with historical context to balance current observations. Models trained on our dataset exhibit substantial improvements in success rates for executing navigation primitives. Our contributions are summarized as follows:

- We systematically identify a critical weakness in end-to-end VLN systems: poor performance on core navigation primitives. On the "move-to" primitive, SOTA models achieve only **43.8%** success rate in MP3D, underscoring a fundamental gap in skills required for long-horizon navigation.

- We develop an automated data generation pipeline that features target acquisition, spatial-semantic enrichment, and MLLM-assisted interactive validation. Finally, construct the **VLMB** dataset focused on the "move-to" primitive. VLMB includes **206** scenarios and **873** object instances, augmented with multi-dimensional spatial-semantic annotations and composable long-horizon instructions. To the best of our knowledge, this is the first step-by-step training dataset specifically designed for end-to-end VLN models.

- We propose a hierarchical memory mechanism incorporating temporal and segment embeddings to balance historical context and current observations. Experiments demonstrate that targeted training on VLMB markedly improves model performance: general-purpose MLLMs improve from **43.8%** to **60.6%** in MP3D and attain **71.4%** in HM3D, affirming the efficacy of our stepwise, primitive-centric approach for enhancing VLN capability.

## 2 RELATED WORK

**Vision-and-Language Navigation (VLN).** The VLN task requires an agent to acquire observational information from the environment and execute specific navigation actions based on natural language instructions. Classical VLN approaches adopt modular architectures (Cai et al., 2024; Yokoyama et al., 2024a; Wanchoo et al., 2024; Chen et al., 2022a; 2024; Yin et al., 2025), decomposing the navigation pipeline into distinct components such as perception, language understanding, mapping, planning, and control. These modules share information to collectively complete the process from instruction interpretation to action execution. However, such systems heavily rely on the construction and alignment of graph structures between scene understanding and language interpretation. They are susceptible to failures if the perception module cannot acquire sufficient scene information, and they suffer from limited inter-module coordination. Furthermore, although advancements in end-to-end network architectures have led to works (Chen et al., 2021; 2022b) that build navigation systems based on deep neural networks or Transformers, these methods, in practice, still depend on external positioning systems (e.g., GPS or SLAM) or multi-sensor inputs (e.g., depth, panoramic images). Therefore, unifying perception and planning is crucial for enhancing decision consistency and generalization, which is vital for the practical deployment of VLN.

**Video-model-based End-to-End VLN.** End-to-end approaches utilizing MLLMs (Zhou et al., 2024; Long et al., 2024; Zheng et al., 2024; Zhang et al., 2025a) present a promising alternative by integrating perception, decision-making, and planning into a unified framework. Given adequate data, MLLMs can directly infer navigation actions from RGB observations and language input, eliminating the need for sensor-based localization or explicit mapping. Existing video-model-based VLN systems (e.g., (Zhang et al., 2024a; Cheng et al., 2025; Wei et al., 2025)) are typically trained

Table 1: A comparative analysis with existing VLN and ObjNav benchmarks.

| Benchmark | Task Type | Simulator | Scenes | Instruction Dimension | Object Categories |
|---|---|---|---|---|---|
| R2R (Anderson et al., 2018) | VLN | Matterport3D | 61 | original goal+ spatial | 21 |
| RxR (Ku et al., 2020) | VLN | Matterport3D | 61 | original goal+ spatial | 21 |
| VLN-CE (Krantz et al., 2020) | VLN | Habitat | 61 | original goal+ spatial | 21 |
| GSA-R2R (Hong et al., 2025) | VLN | Habitat | 150 | original goal+ spatial+ semantic | 21 |
| MP3D ObjectNav (Chang et al., 2017) | ObjNav | Habitat | 61 | original goal | 21 |
| HM3D-OVON (Yokoyama et al., 2024b) | ObjNav | Habitat | 145 | original goal | 379 |
| Goat-Bench (Chang et al., 2023) | ObjNav | Habitat | 145 | original goal+ semantic | 312 |
| VLMB (ours) | VLN+ ObjNav | Habitat | 206 | original goal+ spatial+ semantic | 873 |

and evaluated on long-horizon navigation datasets (Anderson et al., 2018; Ku et al., 2020; Qiao et al., 2025; Gao et al., 2025), which are designed around long trajectories and complex, often ambiguous, instructions. However, these datasets were originally created to evaluate the complex instruction understanding and long-term memory capabilities of modular methods, without considering the functional characteristics of MLLMs. This paper aims to address this critical missing step in training video-based VLN models by introducing the concept of navigation primitives. We propose a comprehensive pipeline to build more stable and effective general-purpose navigation systems, ensuring MLLMs acquire fundamental navigation abilities step-by-step.

# 3 DATASET: VISION-LANGUAGE-MOVE-BASE

## 3.1 PROBLEM SETUP

Given ego-view observations and a language-based navigation instruction, the agent is required to execute a sequence of discrete actions—such as rotating left or right, moving forward, and stopping—to reach a target object or location and halt at the appropriate moment. An episode is deemed successful if the agent stops within a predefined distance threshold from the target (e.g., within a specified radius). A maximum step limit constrains each episode.

Focusing on the "move-to" navigation primitive, we constructed a training set that includes 61 scenes from MP3D (Chang et al., 2017) and 145 scenes from HM3D (Ramakrishnan et al., 2021). For evaluation, we curated an evaluation set consisting of 11 MP3D scenes and 36 previously unseen HM3D scenes. Benefiting from the open semantic annotations provided by HM3D-Semantic, our benchmark covers a richer variety of scenes and a broader range of target instances compared to previous VLN and Object Navigation (ObjNav) datasets, as shown in Table 1.

## 3.2 COLLECTION PROTOCOL

Our data collection pipeline is built upon the Habitat-Sim simulator. Each navigation episode begins from a randomly sampled starting pose, with the target object selected from visually observable entities within the agent's initial field of view. The VLMB construction pipeline consists of three systematically organized stages: (1) an automated target object sampling strategy, (2) a dataset assembly process that incorporates instruction enrichment and interactive validation, and (3) a modular sequential instruction generation pipeline that supports composition operations, as illustrated in Figure 2.

**Stage 1: Automated Goal-Oriented Object Sampling.** In the first stage, the data collection process automatically selects a visible object as the navigation target, while enforcing constraints to ensure that the target is both meaningful and navigable. The target must lie within a distance range of 3–10 meters and be reachable on the Navmap in Habitat-Sim. It must also be visually and semantically distinguishable from surrounding objects to avoid ambiguity. It must belong to a category relevant to navigation, excluding semantically uninformative classes such as *floor* or *wall*. The simulator's path planner validates all selected targets to confirm reachability within the navigation graph, and automated checks are applied to eliminate scenarios prone to collisions or excessive path lengths. The specific rules are described in Appendix A.1.

**Stage 2: Instruction Enrichment and dataset Construction.** In the second stage, we capture the first-frame observation of each episode and employ a high-performing proprietary MLLM

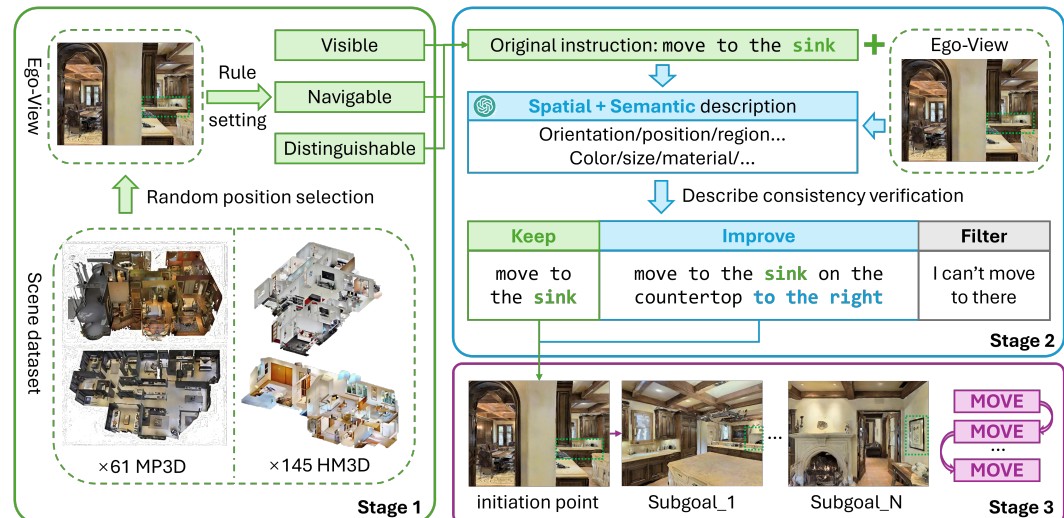

Figure 2: The pipeline of the VLMB dataset generation.

(ChatGPT-4o) to verify the visibility and appropriateness of the selected target. Simultaneously, we expand the base "move-to" instruction with spatial relations and semantic descriptions. Examples include: "move to the chair to the left of the table" and "move to the white sofa in the center of the living room." This process not only enhances linguistic diversity and enriches the complexity of descriptive information, but also cross-validates original and multi-dimensional instructions to filter out targets that are either trivially reachable or highly ambiguous. The curated instruction–trajectory pairs obtained through this stage constitute the VLMB dataset, specifically designed to train the fundamental "move-to" navigation primitive. The specific prompt designs are described in Appendix A.2. The examples of spatial-semantic augmentation and filtering are presented in Figure 6 and 7.

**Stage 3: Structured Multi-Step Instruction Generation.** Building upon the stable execution of navigation primitives, we further collect temporally structured instructions by acquiring new starting points and RGB observations at the end of each navigation segment. Following the procedures of Stage 1 and Stage 2, we obtain multi-step sequential navigation instructions. This demonstrates the advantage of decomposing VLN instructions into fundamental navigation primitives for data collection: the process is highly controllable, and temporal instructions can be easily composed and decomposed. This implies that we can dynamically acquire and execute multi-step instructions with long-horizon extensibility based on practical demands, thereby enabling real-time agent–environment interaction and supporting the construction of complex navigation tasks.

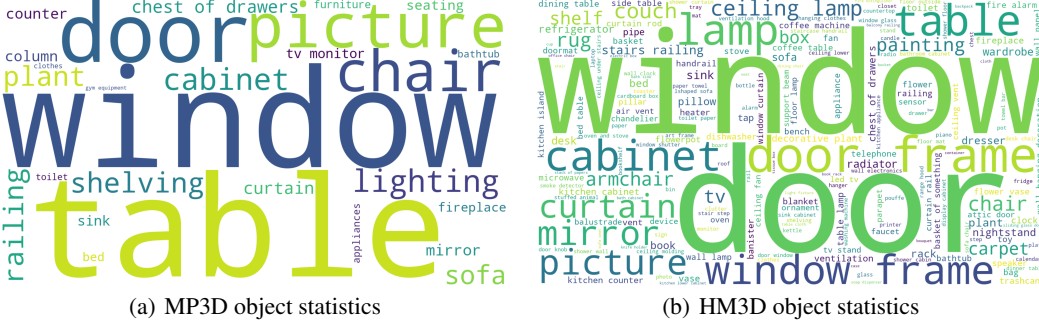

(a) MP3D object statistics      (b) HM3D object statistics

Figure 3: Object Instance Statistics in MP3D and HM3D.

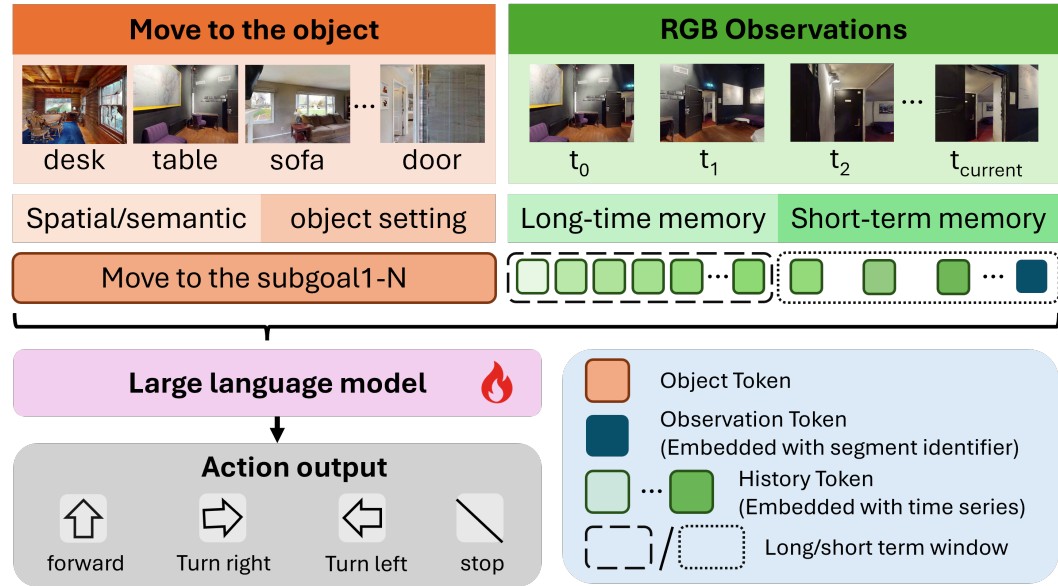

Figure 4: Move-to-Anything framework: architectural overview and modular components.

### 3.3 DATA FORMAT AND ANALYSIS

We collected 30K data samples and retained 15K samples for training after filtering. The word cloud of object categories appearing in MP3D and HM3D is illustrated in Figure 3. Each sample contains a R-Nav instruction (Move to the object) and a VLMB instruction enriched with spatial and semantic descriptions. We randomly sampled a subset of the improved data for manual verification, confirming that the constructed VLMB dataset exhibits high annotation quality. To ensure diversity and coverage, we perform balanced sampling across a wide range of houses and room types, while minimizing the repetition of goals and instructions within the same environment. The details of the dataset construction are provided in Appendix A.3.

To ensure a fair and reliable evaluation of model performance, we construct a dedicated benchmark comprising unseen scenes. Additionally, multi-step instruction compositions are exclusively included in the evaluation set. Beyond the three-stage construction strategy described above, we further apply rigorous manual filtering on top of large model-based selection to guarantee the quality of the evaluation set. These settings enable a thorough assessment of the model's generalization ability in new, unfamiliar objects and paraphrased instructions. Detailed analysis is provided in Appendix B.

## 4 METHOD: MOVE-TO-ANYTHING

Built on LLaVA-Video (Zhang et al., 2024c), which employs Qwen2 (Team, 2024) as the underlying language model, our approach formulates the agent as a multimodal policy that jointly encodes visual observations and language instructions within a unified autoregressive framework to predict discrete actions (e.g., turn, move forward, stop). We train the model using the R-Nav and VLMB datasets constructed in Section 3.

Within this unified pipeline, we introduce two lightweight yet practical temporal memory enhancements to address the high demand for historical context in navigation tasks: (1) a Hierarchical Memory module that preserves critical temporal context while strictly controlling sequence length, and (2) Temporal & Segment Embeddings layered atop the hierarchical memory to explicitly encode recency and memory source, thereby stabilizing long-horizon reasoning and attention allocation. The overall architecture of the Move-to-Anything framework is illustrated in Figure 4.

### 4.1 Hierarchical Memory

To balance recent detail fidelity with global context in long-horizon reasoning, we introduce a hierarchical memory mechanism that separates historical observations into short-term (ST) and long-term (LT) components. Recent frames are encoded by a frozen vision tower, lightly pooled in 2D, and retained as high-fidelity multi-token representations, which are injected into the language model to capture immediate environmental changes. Earlier frames are globally pooled into per-frame features and aggregated into a fixed number of memory slots using a lightweight mean pooling followed by an MLP, achieving an effective trade-off between context preservation and computational efficiency.

This design integrates seamlessly with placeholder-based prompting: ST tokens are inserted as `<image>` and LT summaries as `<history>`. Prefix caching and sliding-window updates further enable efficient streaming inference. By explicitly decoupling ST and LT, token complexity is reduced from $O(T \times S)$ to $O(W \times S + M)$, where $T$ is the total number of frames, $S$ the spatial token count, $W$ the short-term window size, and $M$ the number of long-term memory slots. This significantly lowers computational and memory costs without sacrificing recent detail, improving stability and throughput for long-context reasoning.

### 4.2 Temporal & Segment Embeddings

Building upon the hierarchical memory structure, we introduce two lightweight and learnable embeddings—temporal and segment embeddings—that do not increase the sequence length. For each visual token $x$ from frame $t$ and segment $s$ (either ST or LT), we construct:

$$\tilde{x} = x + E_{\text{time}}[t] + E_{\text{seg}}[s].$$

Here, the temporal embedding $E_{\text{time}}[t]$ encodes the frame index to provide explicit recency cues. In contrast, the segment embedding $E_{\text{seg}}[s]$ identifies the token's origin, distinguishing current observations from aggregated memory. These embeddings are added to visual tokens before fusion with text tokens under the same autoregressive objective. Visual tokens are masked using `IGNORE`, requiring no modification to the training loss.

This explicit conditioning improves attention allocation by prioritizing current evidence before consulting memory, and reduces cross-segment interference, thereby stabilizing long-horizon reasoning. The approach introduces negligible computational overhead while significantly enhancing temporal grounding and memory retrieval efficiency. Moreover, it remains fully compatible with the minimal multimodal injection design of MLLMs.

## 5 Experiments

### 5.1 Settings

**Scene Assets and Evaluation Benchmark.** We construct our experimental environment using the MP3D (Chang et al., 2017) and HM3D (Ramakrishnan et al., 2021) datasets, which together comprise 206 training scenes and 47 unseen evaluation scenes, encompassing over 3,000 rooms with diverse layouts, including kitchens, living rooms, and offices. Semantic annotations were collected for 873 object instances, including doors, table lamps, and televisions. For evaluation, we sample 1,000 instances each from unseen MP3D and HM3D scenes. All evaluations are conducted in Habitat-Sim using R-Nav and VLMB instructions, and are measured with standard VLN metrics: Success Rate (SR), Success weighted by Path Length (SPL), Oracle Success Rate (OS), and Navigation Error (NE).

**Training and Evaluation Settings.** We build the Move-to-Anything framework on top of the LLaVA-Video-7B model (Zhang et al., 2024c), which utilizes Qwen2-7B (Team, 2024) as the language model and adopts the ViT-based visual encoder from SigLIP (Zhai et al., 2023). The visual encoder remains frozen during training. For optimization, we employ the Adam optimizer with a learning rate of $3 \times 10^{-5}$. In the simulation, we use the Stretch robot from Hello Robot. Stretch is equipped with a wheeled base and a manipulator mounted on a structural frame, and it executes four discrete actions: turn left, turn right, move forward, and stop. The robot is fitted with an RGB camera to capture front-view images.

Table 2: Comparison with previous methods on R-Nav and VLMB. Best results are in **bold**. NaVid was not included in the HM3D comparison because it was not trained on the HM3D environment.

| Method | R-Nav | | | | | | | | VLMB | | | | | | | |
|---|---|---|---|---|---|---|---|---|---|---|---|---|---|---|---|---|
| | MP3D | | | | HM3D | | | | MP3D | | | | HM3D | | | |
| | SR↑ | SPL↑ | OS↑ | NE↓ | SR↑ | SPL↑ | OS↑ | NE↓ | SR↑ | SPL↑ | OS↑ | NE↓ | SR↑ | SPL↑ | OS↑ | NE↓ |
| *MLLMs as Agent* | | | | | | | | | | | | | | | | |
| Vedio-LLaVA (Lin et al., 2024) | 14.1 | 12.5 | 34.7 | 4.91 | 15.2 | 14.6 | 43.1 | 5.02 | 13.4 | 12.3 | 34.3 | 4.96 | 15.3 | 14.1 | 39.2 | 5.21 |
| LLaVA-NEXT (Zhang et al., 2024b) | 12.1 | 10.5 | 31.5 | 5.46 | 14.2 | 13.5 | 39.4 | 5.45 | 11.2 | 10.5 | 33.1 | 5.34 | 12.3 | 11.5 | 35.7 | 5.56 |
| GPT-4o | 11.6 | 11.4 | 13.6 | 5.23 | 22.6 | 22.5 | 24.8 | 4.53 | 15.3 | 15.2 | 16.9 | 4.98 | 20.29 | 20.29 | 20.45 | 4.44 |
| GPT-5-mini | 14.1 | 14.1 | 14.2 | 5.02 | 25.4 | 25.3 | 27.1 | 4.29 | 16.1 | 16.1 | 17.1 | 4.92 | 25.77 | 25.77 | 26.5 | 4.36 |
| *Method for VLN* | | | | | | | | | | | | | | | | |
| NaVid (Zhang et al., 2024a) | 46.7 | 42.6 | 59.8 | 4.74 | – | – | – | – | 43.8 | 39.0 | 60.1 | 4.89 | – | – | – | – |
| StreamVLN (Wei et al., 2025) | 15.8 | 10.8 | 52.2 | 8.48 | 32.4 | 28.2 | 49.3 | 5.43 | 22.0 | 16.0 | 60.9 | 7.24 | 44.2 | 39.0 | 59.3 | 4.43 |
| **Move-to-Anything (ours)** | **62.5** | **61.4** | **69.1** | **3.09** | **70.6** | **69.6** | **75.8** | **2.70** | **60.6** | **59.6** | **67.6** | **3.21** | **71.4** | **70.3** | **76.3** | **2.71** |

Table 3: Comparison with previous methods on multi-step navigation task.

| Method | R-Nav | | | | VLMB | | | |
|---|---|---|---|---|---|---|---|---|
| | SR↑ | SPL↑ | OS↑ | NE↓ | SR↑ | SPL↑ | OS↑ | NE↓ |
| StreamVLN (Wei et al., 2025) | 11.5 | 2.36 | 50.4 | 10.3 | 20.4 | 5.25 | 61.9 | 10.26 |
| NaVid (Zhang et al., 2024a) | 31.9 | 28.5 | 44.2 | 7.24 | 32.4 | 28.1 | 47.8 | 6.45 |
| **Move-to-Anything (ours)** | **36.3** | **29.1** | **54.9** | **6.24** | **35.4** | **28.7** | **53.1** | **6.38** |

## 5.2 MAIN RESULTS

We systematically evaluate general-purpose MLLMs, representative end-to-end VLN models, and the proposed method on single-step action execution tasks using the R-Nav and VLMB benchmarks, as shown in Table 2. It is noteworthy that within the "MLLMs as Agent" framework, four 7B-parameter base models and a closed-source GPT model were evaluated, all of which demonstrated suboptimal performance on the test set. This finding further corroborates our hypothesis that existing training-free base models lack fundamental navigation capabilities, and directly training them on long-horizon navigation datasets is not justified. **Our Move-to-Anything approach achieves SOTA performance on both benchmarks, demonstrating the effectiveness of the VLMB dataset in enhancing foundational navigation abilities.**

To further validate the potential of Move-to-Anything in long-horizon tasks, we constructed a multi-step instruction evaluation set by composing single-step actions from VLMB using the method described in Section 3.2, Stage 3, as shown in Table 3. **Notably, our model outperforms existing SOTA models, despite never being trained on long-horizon instruction samples.** In contrast, most current SOTA models are trained on datasets specifically designed around long-horizon instruction paradigms. In contrast, most current SOTA models are trained on datasets specifically designed around long-horizon instruction paradigms.

## 5.3 VISUALIZATIONS

We recorded and visualized selected evaluation samples, as shown in Figure 5 and Appendix C. The results show that Move-to-Anything generalizes well to unseen environments and can execute VLN tasks based on navigation primitives. To illustrate this, we selected three representative VLMB instructions. The first captures spatial relationships among objects within a room. The second demonstrates precise localization of the target object, even in the presence of similar distractors and obstacles. The third highlights the model's ability to utilize new semantic cues, such as the "glowing exit sign", which were not included in the training set, but improved the accuracy of the target description.

We also demonstrate the model's ability to complete complex tasks using multi-step instructions. The agent's actual navigation trajectory does not always match the simulator's shortest path, as the stopping point for receiving new instructions may not coincide with the predefined collection point. Nevertheless, the agent can still complete navigation tasks based on RGB observations and instructions. This suggests that models trained with navigation primitives exhibit enhanced scene understanding and planning capabilities. In contrast, traditional instruction paradigms often struggle when the agent's observations do not match the instructions.

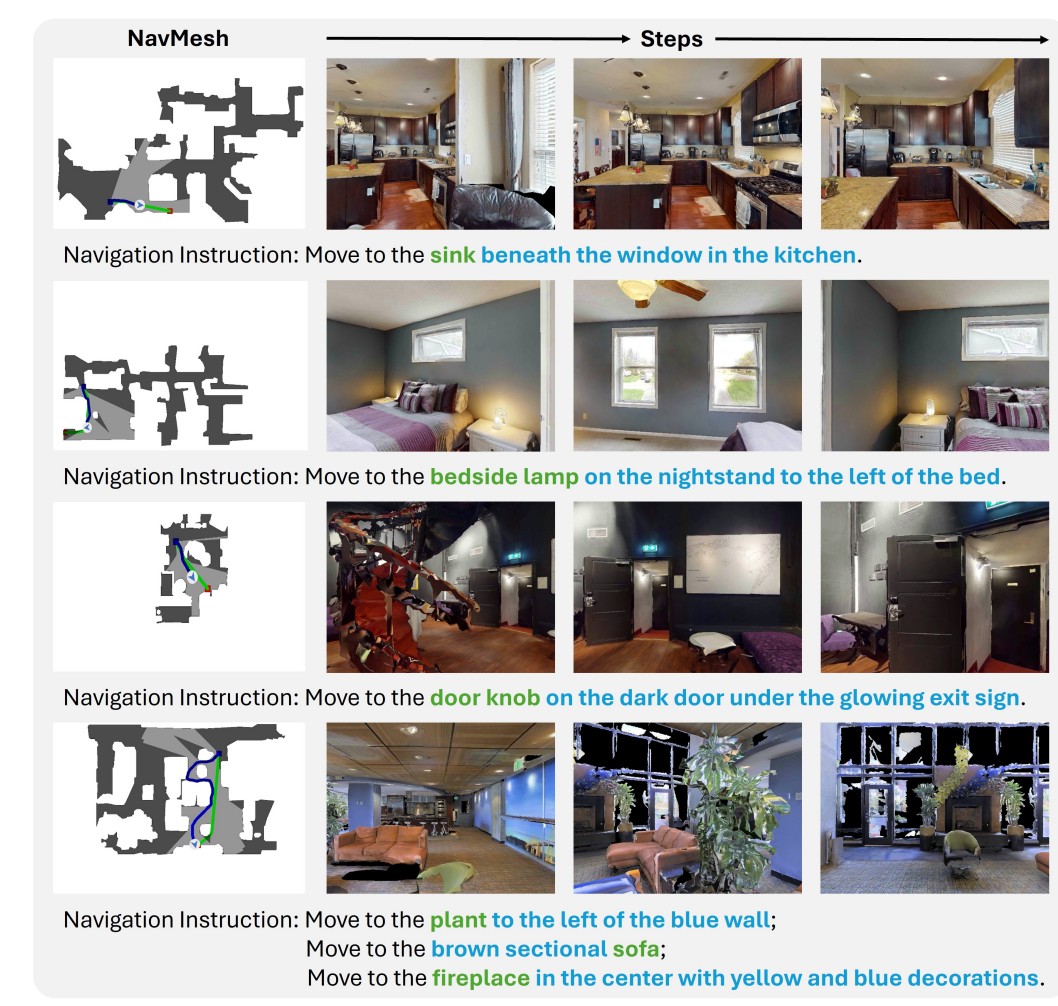

Figure 5: Examples of visualized trajectories and observed images. In the navigation mesh (NavMesh), the color green denotes the reference path, while blue indicates the actual path taken by the agent.

Table 4: Comparison with SOTA methods on VLN-CE R2R Val-Unseen split.

| Method | Traning Dataset | External Data | R2R Val-Unseen | | | |
|---|---|---|---|---|---|---|
| | | | SR↑ | SPL↑ | OS↑ | NE↓ |
| NaVid (Zhang et al., 2024a) | R2R+RxR+VLNdata | 953K | 37.4 | 25.9 | 49.1 | 5.47 |
| StreamVLN (Wei et al., 2025) | R2R+RxR+EnvDrop | 10033K | 45.5 | 41.6 | 53.8 | 6.05 |
| Move-to-Anything* (ours) | R2R+RxR | 0k | 33.6 | 30.1 | 43.1 | 7.00 |
| Move-to-Anything (ours) | R2R+RxR+VLMB | 40K | 38.0 (**+4.4**) | 34.5 (**+4.4**) | 43.2 (**+0.1**) | 6.45 (**-0.55**) |

## 5.4 COMPARED IN R2R

We have verified that the compact dataset proposed in this paper effectively enhances the model's fundamental navigation skills. To further validate the contribution of the VLMB dataset to navigation capability, we conducted joint training using VLMB together with R2R and RxR datasets—even though we maintain that the instruction composition of R2R alone is insufficient for equipping models with general navigation competence. **Experimental results show that by incorporating even a small amount of VLMB data during joint training, the model achieves higher performance levels on R2R, demonstrating the usefulness and effectiveness of the VLMB dataset.** In Appendix B.2, we present a detailed comparison of the model's training duration and inference time.

Table 5: Ablation studies on dataset cross-validation and model improvements.

| Method | R-Nav | | | | | | | | VLMB | | | | | | | |
|---|---|---|---|---|---|---|---|---|---|---|---|---|---|---|---|---|
| | MP3D | | | | HM3D | | | | MP3D | | | | HM3D | | | |
| | SR↑ | SPL↑ | OS↑ | NE↓ | SR↑ | SPL↑ | OS↑ | NE↓ | SR↑ | SPL↑ | OS↑ | NE↓ | SR↑ | SPL↑ | OS↑ | NE↓ |
| w/o cross validation of datasets | 55.7 | 54.0 | 64.6 | 3.45 | 67.3 | 66.0 | 75.3 | 2.89 | 54.0 | 52.6 | 63.1 | 3.53 | 67.8 | 66.2 | 75.6 | 2.84 |
| w/o HM and TSE | 57.6 | 56.5 | 67.1 | 3.43 | 66.3 | 65.3 | 73.7 | 2.84 | 58.9 | 57.7 | 67.1 | 3.32 | 66.0 | 64.9 | 73.3 | 2.86 |
| w/o TSE | 59.4 | 58.1 | 68.3 | 3.34 | 68.2 | 68.4 | 74.2 | 2.89 | 59.3 | 58.3 | 67.5 | 3.25 | 68.1 | 66.5 | 74.1 | 2.80 |
| **Move-to-Anything (ours)** | **62.5** | **61.4** | **69.1** | **3.09** | **70.6** | **69.6** | **75.8** | **2.70** | **60.6** | **59.6** | **67.6** | **3.21** | **71.4** | **70.3** | **76.3** | **2.71** |

Table 6: Ablation study on HM configuration: impact of short-term window size ($W$), long-term memory slots ($M$), and aggregation method.

| W | M | Aggregator Type | VLMB | | | | | | | |
|---|---|---|---|---|---|---|---|---|---|---|
| | | | MP3D | | | | HM3D | | | |
| | | | SR↑ | SPL↑ | OS↑ | NE↓ | SR↑ | SPL↑ | OS↑ | NE↓ |
| 4 | 784 | | 53.2 | 51.8 | 60.3 | 3.78 | 64.7 | 63.1 | 69.2 | 3.12 |
| 8 | 784 | | 57.1 | 55.7 | 64.4 | 3.45 | 68.9 | 67.5 | 73.4 | 2.89 |
| 8 | 1,568 | Mean+MLP | **60.6** | **59.6** | **67.6** | **3.21** | **71.4** | **70.3** | **76.3** | **2.71** |
| 16 | 1,568 | | 58.8 | 57.9 | 65.9 | 3.38 | 70.6 | 69.4 | 75.1 | 2.74 |
| 8 | 1,568 | N/A | 57.7 | 56.5 | 64.2 | 3.40 | 68.9 | 67.6 | 73.8 | 3.01 |

## 5.5 ABLATION STUDIES

To validate the effectiveness of our dataset construction and model improvements, we conducted ablation studies as shown in Table 5. In the initial stage of automated goal-oriented object sampling for VLMB, we collected 30,000 samples. Through rigorous cross-validation and filtering with a large model, we retained 15,000 high-quality samples. **This strategy not only halved the dataset size but also led to a significant improvement in model performance, demonstrating the value of our data selection process.** Furthermore, the two lightweight modules proposed in this work, hierarchical memory (HM) and temporal & segment embeddings (TSE), demonstrated significant effectiveness in ablation studies. These modules enhance the model's ability to capture temporal dependencies, thereby improving its adaptability to navigation tasks.

We performed ablation experiments on the parameter settings within the HM module. Table 6 reports results when varying the short-term window size and the number of long-term memory slots, and compares our default mean+MLP aggregation strategy with a no-aggregation baseline. **Results show that increasing both $W$ and $M$ improves navigation performance up to an optimal configuration ($W=8$, $M=1568$), while the mean+MLP aggregator achieves a favorable balance between hierarchical memory preservation and computational efficiency.**

## 6 CONCLUSION

This work identifies a key weakness in current end-to-end VLN systems: although they perform well on long-horizon navigation, they struggle with basic navigation primitives. To this end, we introduce VLMB, the first dataset specifically designed to enhance the fundamental navigation capabilities of end-to-end VLN systems. Centered around the "move-to" navigation primitive, VLMB constructs a complete training set and evaluation benchmark through MLLM-based spatial and semantic augmentation, enabling controllable composition of these primitives. Experiments show that our proposed VLMB dataset and hierarchical memory design offer a principled approach to equipping general-purpose MLLMs with reliable navigation capabilities. Overall, we present a scalable, step-by-step training paradigm that helps general-purpose MLLMs acquire reliable foundational skills for VLN applications.

## ETHICS STATEMENT

This work does not involve human subjects, sensitive personal data, or practices that raise privacy or security concerns. We have taken care to avoid introducing or amplifying bias or discrimination in our models and datasets. No conflicts of interest or sponsorship-related influences have affected

the research process or outcomes. We remain committed to upholding ethical standards throughout the submission, review, and discussion phases of the ICLR conference.

## REPRODUCIBILITY STATEMENT

We are committed to the transparency and reproducibility of our findings. The code and datasets will be made publicly available to the research community, enabling others to reproduce, verify, and build upon our work. Additionally, to ensure reproducibility, we have also provided detailed descriptions of the dataset configuration pipeline in the Appendix. The link to the anonymous repository is as follows: `https://anonymous.4open.science/r/move_to_anything-4E1F`.

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

## A  DATASET CONSTRUCTION

### A.1  RULE SETTING

Let $S$ denote the semantic segmentation observation, $p_a$ the current agent position, and $\mathcal{O}$ the scene objects indexed by semantic IDs. We first form the set of visible IDs $\mathcal{I} = \mathrm{unique}(S) \setminus \{0\}$ and define the candidate set

$$\mathcal{V} = \Big\{ o \in \mathcal{O} \,\Big|\, \mathrm{id}(o) \in \mathcal{I},\ \mathrm{cat}(o) \notin \mathcal{U},\ \mathrm{cat}(o) \neq \varnothing,\ d(o) = \mathrm{dist}_g(p_a, p_o) < \infty \Big\},$$

where $\mathcal{U}$ is the excluded category set (e.g., wall, floor, ceiling), $p_o$ is the OBB center of $o$, and $\mathrm{dist}_g$ is the simulator geodesic distance. We retain objects whose category occurs exactly once in view,

$$\mathcal{V}_{\mathrm{uniq}} = \Big\{ o \in \mathcal{V} \,\Big|\, |\{o' \in \mathcal{V} : \mathrm{cat}(o') = \mathrm{cat}(o)\}| = 1 \Big\},$$

and apply a distance filter

$$\mathcal{V}_d = \{\, o \in \mathcal{V}_{\mathrm{uniq}} \mid 3.0 \leq d(o) \leq 10.0 \,\}.$$

If $\mathcal{V}_d \neq \varnothing$, we sample $o^* \sim \mathrm{Unif}(\mathcal{V}_d)$, snap its position to the navigation mesh $p^* = \mathrm{snap}(p_o^*)$, and abort if snapping fails. A shortest-path follower with goal radius $r = 3\,\mathrm{m}$ and step cap $T = 150$ is used to navigate from the current pose to $p^*$. Success is declared iff there exists $t \in \{1, \ldots, T\}$ such that $\mathrm{dist}_g(p_t, p^*) \leq r$, where $p_t$ is the agent position at step $t$. Upon success, we persist the sample with the command "move to the $\mathrm{cat}(o^*)$", a fixed-length (100) placeholder token sequence, the reference path, and navigation metadata; otherwise, the sample is discarded. For a chained collection with $N$ segments (no reset), the procedure is repeated per segment; segment commands are concatenated with semicolons, reference paths are concatenated, geodesic distances are summed, and the first RGB frame of each segment is recorded. The episode is valid only if all segments succeed.

### A.2  PROMPT DEFINITIONS

SYSTEM PROMPT

> You are a precise instruction rewriter for a mobile robot. You receive: (1) a base movement instruction string, (2) an image. Output exactly ONE short, executable movement command that keeps the same target and intent as the base instruction, and, when possible, enrich it using ONLY details clearly observed in the image.
>
> **Hard rules:**
>
> - The final command must remain aligned with the original target (do not change what to move).
> - Do NOT invent or guess details that are not visible in the image.
> - Avoid uncertainty words (e.g., maybe, probably, around, approximately, seems).
> - Output exactly one sentence, no explanations, no lists.
> - If the target object is NOT visible anywhere in the image, output exactly: `I can't move to there`
> - If the target object IS visible but you cannot confidently add details, output the base instruction unchanged.
>
> **Description styles (choose based on what is visible):**
>
> - **Spatial relation:** use clear relative positions with salient anchors (e.g., left/right/front/behind: "move to the chair to the left of the table").
> - **Semantic attributes:** use clear attributes like color/material/type (e.g., "move to the green sofa").
>
> Pick the style that yields the most explicit single-sentence command; if both are obvious and concise, you may combine them briefly.

USER PROMPT (EXAMPLE)

Base instruction: move to the table

Using ONLY what is visible in the image, produce ONE precise, executable command that preserves the same target. Prefer either a spatial-relation description or a semantic-attribute description based on what is clearly visible. If the target object is not visible at all, reply exactly: `i cant move to there`. If the target object is visible but you cannot confidently add details, repeat the base instruction unchanged.

### A.3  DATASET CONSTRUCTION PIPELINE

Through a two-stage sampling and optimization process, we can automatically construct the VLMB dataset. Figure 6 shows the instruction information and the first-frame RGB observation image after adding semantic and spatial descriptions. It can be observed that the large model enriches the original "move to the object" command by adding a series of details such as orientation, facing direction, color, and material. The descriptive dimensions for the target are expanded, and the relationships between objects are also introduced, thereby enhancing the model's understanding of the scene. This, in turn, improves its ability to localize the target and boosts overall navigation performance. This pipeline also effectively filters out incorrectly labeled images from the collector, as shown in Figure 7. In addition to inherent annotation errors from the simulator, these images include unreachable targets such as ceiling lamps, as well as objects that are difficult to identify in the observation images. This step helps prevent the generation of erroneous training data. Similarly, we apply the same filtering strategy to the evaluation data to ensure the validity of the evaluation.

## B  COMPARISON WITH EXISTING METHOD

### B.1  COMPARISON WITH EXISTING VLN DATASET

Most existing VLN datasets are derived from VLN-CE (R2R and RxR), with model training and evaluation primarily conducted on these corpora. We categorize R2R/RxR instructions into four common navigation primitives based on their functional semantics: `move` (direct displacement without turning or crossing region boundaries), `situation` (pure contextual descriptions without explicit action requests), `change region` (crossing spatial boundaries or switching areas, e.g., entering/leaving rooms or floors), and `change direction` (orientation changes such as turning left/right without necessary displacement). Although VLN-CE instructions are often verbose and compositional, their clauses can be reliably mapped to these four types. Consequently, R-Nav (move-to-object) is extensively represented within VLN-CE, functioning as its subset and effectively serving as a structural simplification of VLN-CE's complex instructions. Automated classification using GPT-4o, and manual verification, we obtained the proportions presented in Table 7, where `move` accounts for the largest share. Importantly, `move-to` (our operationalization of `move`) is closest to human navigation phrasing and is the easiest to construct at scale. By semantically and spatially extending `move-to`, we subsume most `situation` statements and `change region` transitions, and optionally pair with minimal `change direction` tokens when orientation is required. For example, a `change region` instruction such as "*walk towards the bathroom*" can be expressed using `move-to`, and a `situation` instruction like "*wait in the doorway of the dining room*" can similarly be represented within the `move-to` framework. Building on these observations, we constructed VLMB.

**Limitations of VLN-CE**

- **Limited scene diversity and semantics.** Both datasets are confined to MP3D with only 61 training scenes focused on indoor home settings; semantic annotations cover merely 21 categories, constraining generalization of end-to-end models.

- **Lack of foundational capabilities.** Classical modular systems (with maps/localization) can reliably reach targets within the field of view, whereas general end-to-end models typically cannot. The community often overlooks that stable path planning is more fundamental than instruction following, leading to brittle behaviors.

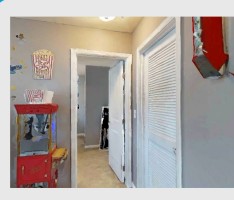 **Original** instruction:
Move to the mirror
**Improved** instruction:
Move to the mirror visible
through the doorway
ahead

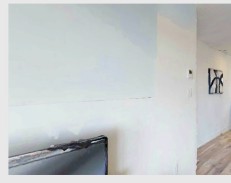 **Original** instruction:
Move to the painting
**Improved** instruction:
Move to the painting with
black abstract lines on the
right wall

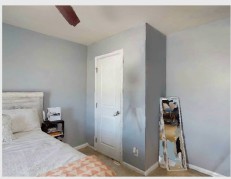 **Original** instruction:
Move to the bedside lamp
**Improved** instruction:
Move to the bedside lamp
on the table next to the
bed

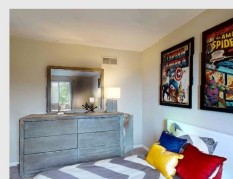 **Original** instruction:
Move to the lamp
**Improved** instruction:
Move to the lamp on the
dresser

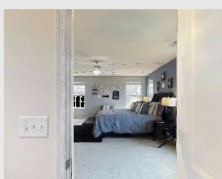 **Original** instruction:
Move to the bed table
**Improved** instruction:
Move to the table to the
right of the bed

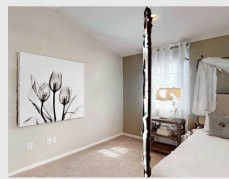 **Original** instruction:
Move to the curtain
**Improved** instruction:
Move to the curtain behind
the bedside table

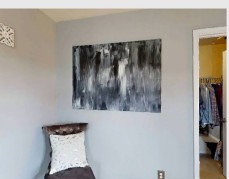 **Original** instruction:
Move to the clothes
**Improved** instruction:
Move to the clothes inside
the open closet to the
right of the painting

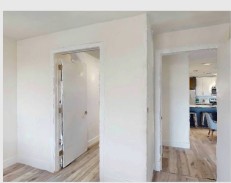 **Original** instruction:
Move to the worktop
**Improved** instruction:
Move to the worktop
visible through the right
doorway in the kitchen

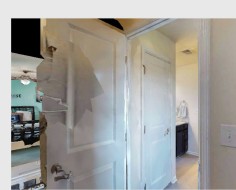 **Original** instruction:
Move to the sink
**Improved** instruction:
Move to the sink visible on
the black counter to the
right

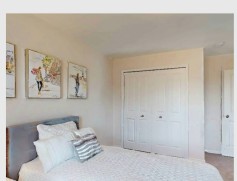 **Original** instruction:
Move to the door frame
**Improved** instruction:
Move to the door frame on
the right

Figure 6: Examples of refined instructions and RGB observations after semantic and spatial augmentation.

- **Unstructured instruction design.** Current instructions do not support step-wise execution or timely intervention, obscuring failure points and hindering robust control for complex tasks.

**VLMB: Targeted Improvements**

- **Enhanced scene and semantic diversity.** VLMB includes 206 training scenes and 873 instances; with large-model annotations, the effective semantic categories and spatial relations are substantially richer.

- **Primitive-level analysis and training.** We find that 42% of R2R instructions can be abstracted into `move`. We explicitly train/evaluate this capability, which—though seemingly simple—requires target understanding/localization, path planning, and stop decisions.

- **Multi-step, controllable instruction construction.** By enriching `move` and composing multi-step commands, VLMB supports long-range controllable navigation with stage-wise monitoring and intervention, enabling recovery after localized failures.

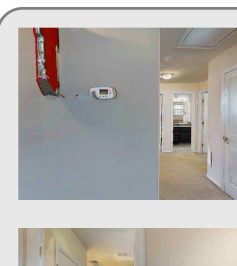
Original instruction:
Move to the chandelier
Filtered instruction:
I can't move to there

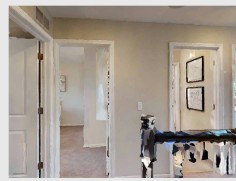
Original instruction:
Move to the sink cabinet
Filtered instruction:
I can't move to there

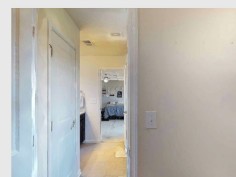
Original instruction:
Move to the bath cabinet
Filtered instruction:
I can't move to there

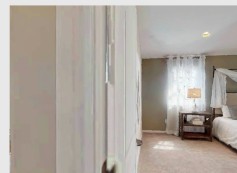
Original instruction:
Move to the balustrade
Filtered instruction:
I can't move to there

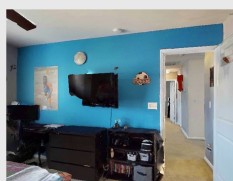
Original instruction:
Move to the mirror
Filtered instruction:
I can't move to there

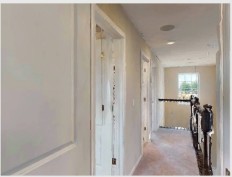
Original instruction:
Move to the picture
Filtered instruction:
I can't move to there

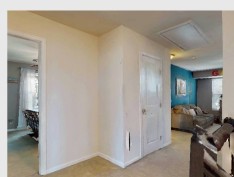
Original instruction:
Move to the ceiling lamp
Filtered instruction:
I can't move to there

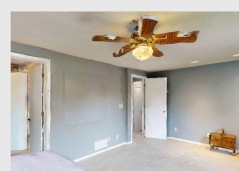
Original instruction:
Move to the smoke detector
Filtered instruction:
I can't move to there

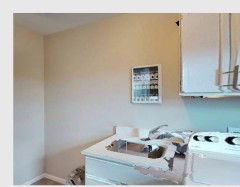
Original instruction:
Move to the bed
Filtered instruction:
I can't move to there

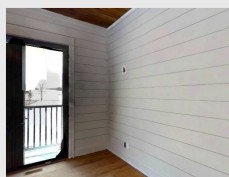
Original instruction:
Move to the table
Filtered instruction:
I can't move to there

Figure 7: Examples of incorrectly labeled images filtered out by the pipeline.

Table 7: Proportions of common navigation primitive types.

| Navigation Primitive | Instruction Numbers | Percentage |
|---|---|---|
| Move | 227,189 | 42% |
| Situation | 200,143 | 37% |
| Change Direction | 70,320 | 13% |
| Change Region | 43,274 | 8% |

## B.2 COMPARISON WITH EXISTING VLN MODEL

We conducted a visual case study of the navigation trajectories and ego-observations of existing SOTA models on the R2R dataset. Our analysis of two successfully labeled cases (Figure 8) reveals critical issues: in the first case, the navigation model initially moves in the complete opposite direction of the instruction, eventually reaching the goal after a circular detour. Although labeled as successful, the model clearly demonstrates navigational disorientation. In the second case, despite explicit instructions to *stop at the first entrance*, the model terminates at an incorrect location. These cases reveal substantial randomness in the model's success criteria, model can reach the target even when completely ignoring instructions from the outset. This fundamentally contradicts the purpose of instruction-following navigation. We attribute these errors to the training paradigm where mod-

Table 8: Training and inference comparison across models.

| Method | Traning dataset | Dataset size | Traning time (GPU hours) | Inference time (s) R2R | VLMB |
|--------|----------------|--------------|--------------------------|------------------------|------|
| NaVid | R2R+RxR+VLNdata | 953K | 672 | 7.9638 | / |
| StreamVLN | R2R+RxR+EnvDrop | 10033K | 1500 | 8.3288 | / |
| Move-to-Anything (ours) | VLMB | 40K | 76 | / | 3.0852 |

els are exposed to repetitive samples with ambiguous instructions, causing MLLMs to overfit to spurious vision-language correlations rather than learning genuine navigation skills.

A quantitative comparison of the training dataset scale, training duration, and inference time between Move-to-Anything and existing SOTA models is presented in Table 8. Both training and inference were conducted on NVIDIA A100 GPUs. The training utilized $4\times80$ GB A100 cards, with a total memory footprint of 285 GB, and the training duration was reported in terms of A100 GPU hours. For inference, the memory consumption was approximately 30 GB, and the reported inference time corresponds to the complete instruction-response cycle. **As evidenced by the results, our Move-to-Anything model requires substantially fewer training samples compared to conventional SOTA approaches. Furthermore, benefiting from its streamlined instruction format, our method achieves significantly faster inference speed than existing models.**

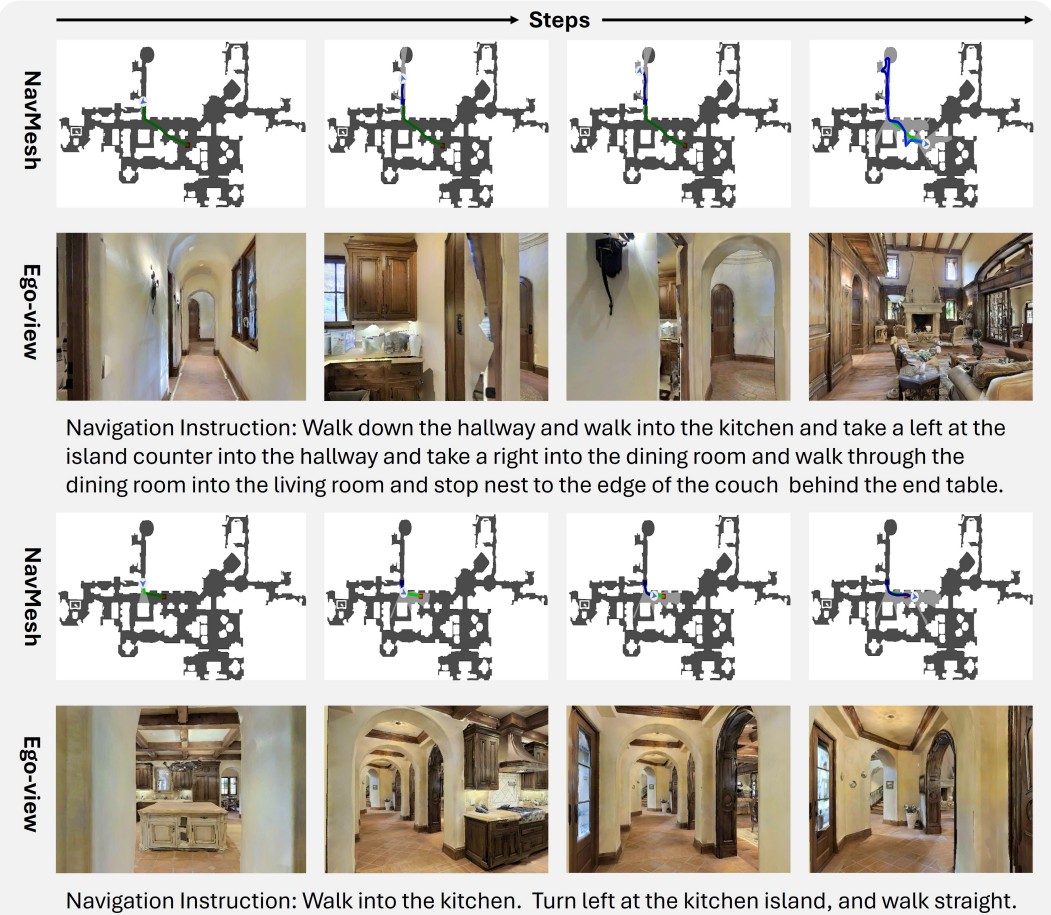

Figure 8: Visualization of successful samples in R2R. In the navigation mesh (NavMesh), the color green denotes the reference path, while blue indicates the actual path taken by the agent.

## C VISUALIZATION RESULTS

We selected a subset of samples from the HM3D scenes for visualization, as shown in Figure 9. These samples include various spatial and semantic characteristics: directional relations (e.g., *on the left side of the room*), regional cues (e.g., *visible through the doorway straight ahead*), objects located at the edge of the observation field (often easily overlooked), and region containment relations (e.g., *in the kitchen*). Together, these examples represent a broad range of spatial and semantic information commonly encountered in navigation tasks.

We further visualize samples from multi-step instruction navigation tasks, as shown in Figure 10. In some cases, a noticeable gap exists between the actual navigation path taken by the agent and the optimal path generated by the simulator. This observation suggests that the model is not merely memorizing predefined routes but instead demonstrates a certain level of path planning capability. These results support the effectiveness of the navigation primitive-based paradigm in building general-purpose navigation systems.

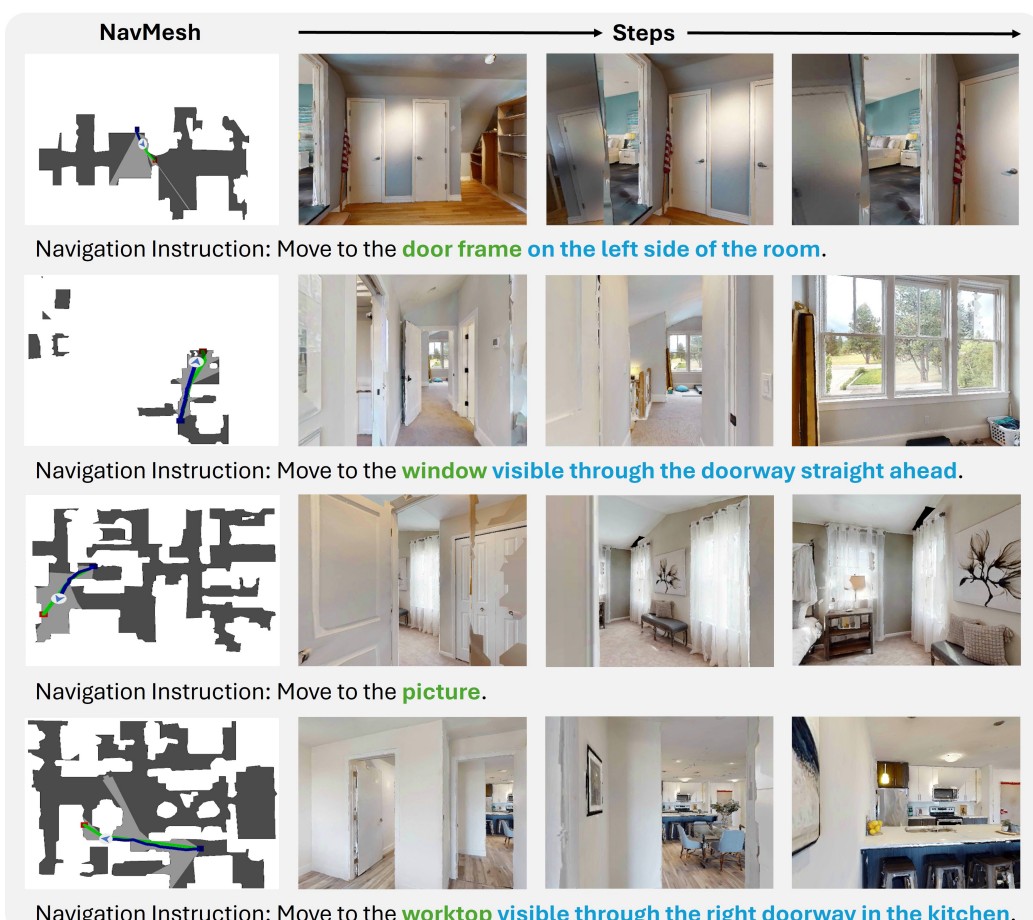

Figure 9: Examples of visualized trajectories and observed images in HM3D.

## D USE OF LARGE LANGUAGE MODELS (LLMS)

We used Large Language Models (LLMs) in two limited ways: (1) ChatGPT was employed to polish the language and improve clarity of the manuscript; (2) During dataset construction, ChatGPT assisted in generating candidate instructions and validating specific annotation steps under human supervision. All outputs were manually reviewed to ensure accuracy and integrity. No experimental results or analyses were produced solely by LLMs.

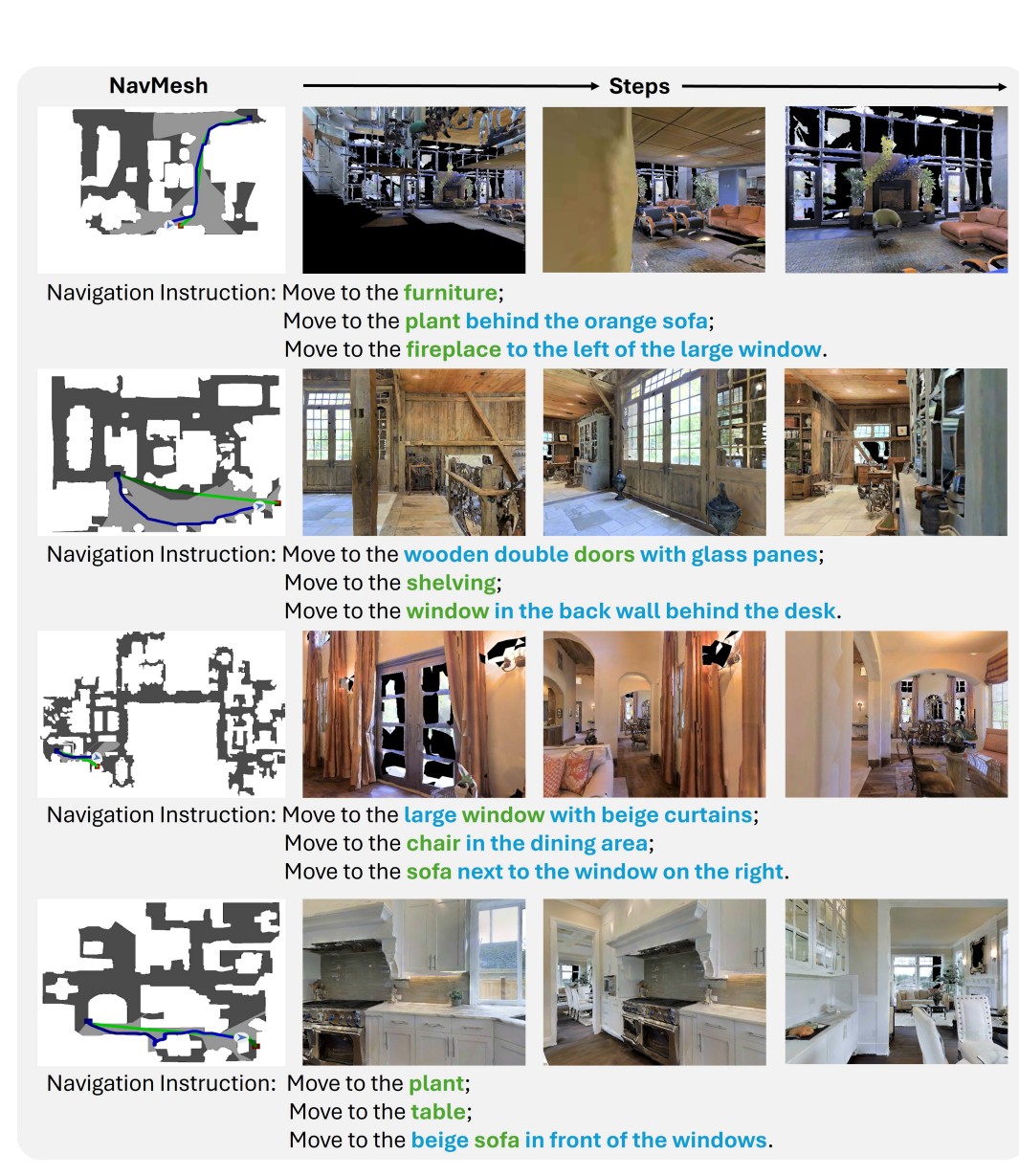

Figure 10: Examples of visualized trajectories and observed images in a multi-step navigation task.

