# Response to Reviewer 1:

We thank the reviewer for the constructive feedback. Below are our responses to the major concerns raised. The corresponding modified sections in the manuscript have been marked in **Blue**.

**Weakness 1: Generalization to dynamic obstacles and extreme weather**

Thank you for raising this important point. We would like to clarify that Video-LLM-based VLN models (e.g., Zhang et al. (2024a); Wei et al. (2025)) primarily target semantically rich indoor environments, whereas dynamic obstacles and extreme weather are more characteristic of outdoor settings with sparse semantics. **The primary goal of VLN research is to enable agents to interpret diverse human instructions and execute navigation actions accurately in environments with dense semantic cues. The core motivation of our work is to address a concerning trend in the VLN community: existing models increasingly prioritize visual-text alignment in long-horizon tasks while overlooking fundamental navigation skills.** To refocus on navigation competence, we designed a comprehensive automated data generation pipeline and trained a model to validate its effectiveness. Importantly, a major part of our motivation is to equip MLLMs with true all-weather and all-terrain navigation capabilities. We fully acknowledge the significance of real-world challenges and plan to extend our framework to more diverse and complex conditions in future work.

**Weakness 2: Ablation studies on Hierarchical Memory**

We have supplemented ablation studies on the short-term window size ($W$) and long-term memory slots ($M$) in Section 5.5 and Table 6. **Results show that increasing both $W$ and $M$ improves navigation performance up to an optimal configuration ($W$=8, $M$=1568), while the mean+MLP aggregator achieves a favorable balance between hierarchical memory preservation and computational efficiency. The reason we did not conduct an isolated ablation for HM (i.e., TSE without HM) is that the Temporal & Segment Embeddings module is designed to depend on HM, as it incorporates temporal and semantic embeddings based on HM's token selection.** The two components are inherently coupled in our design.

Table 1: Ablation study on HM configuration: impact of short-term window size ($W$), long-term memory slots ($M$), and aggregation method.

| W | M | Aggregator Type | VLMB | | | | | | | |
|---|---|---|---|---|---|---|---|---|---|---|
| | | | MP3D | | | | HM3D | | | |
| | | | SR↑ | SPL↑ | OS↑ | NE↓ | SR↑ | SPL↑ | OS↑ | NE↓ |
| 4 | 784 | | 53.2 | 51.8 | 60.3 | 3.78 | 64.7 | 63.1 | 69.2 | 3.12 |
| 8 | 784 | Mean+MLP | 57.1 | 55.7 | 64.4 | 3.45 | 68.9 | 67.5 | 73.4 | 2.89 |
| 8 | 1,568 | Mean+MLP | **60.6** | **59.6** | **67.6** | **3.21** | **71.4** | **70.3** | **76.3** | **2.71** |
| 16 | 1,568 | | 58.8 | 57.9 | 65.9 | 3.38 | 70.6 | 69.4 | 75.1 | 2.74 |
| 8 | 1,568 | N/A | 57.7 | 56.5 | 64.2 | 3.40 | 68.9 | 67.6 | 73.8 | 3.01 |

**Weakness 3: Quantitative analysis of model complexity and efficiency**

Thank you for raising this point. To ensure a fair comparison, our model was trained with the same backbone (LLaVA-Vid) and identical parameter settings as NaVid and StreamVLN. We provide additional comparisons in Appendix B.2 and Table 8, including dataset size, training time, and inference speed. **Results show that our approach requires significantly less training data, reducing training cost, and achieves faster inference thanks to the streamlined instructions in VLMB.**

Table 2: Training and inference comparison across models.

| Method | Traning dataset | Dataset size | Traning time (GPU hours) | Inference time (s) | |
|---|---|---|---|---|---|
| | | | | R2R | VLMB |
| NaVid | R2R+RxR+VLNdata | 953K | 672 | 7.9638 | / |
| StreamVLN | R2R+RxR+EnvDrop | 10033K | 1500 | 8.3288 | / |
| Move-to-Anything (ours) | VLMB | 40K | 76 | / | 3.0852 |

We sincerely appreciate the reviewer's insightful comments, which have been instrumental in enhancing the quality of our manuscript.

## Response to Reviewer 2:

We sincerely thank the reviewer for their insightful comments and constructive feedback. We have carefully revised the manuscript to address the concerns raised. Our point-by-point responses are detailed below. The corresponding modified sections in the main text have been marked in **Red**.

**Weakness 1: Clarification of Motivation and Comparison with Alternative Approaches**

**Clarification of Motivation.** We acknowledge that the initial motivation presented in the paper may have been insufficiently elaborated. Our core argument is that the current training paradigm for end-to-end VLN models, particularly those based on Video-LLM-based frameworks, is predominantly evaluated on similar datasets (e.g., R2R, RxR). **This narrow focus has led models to overemphasize the alignment between visual inputs and lengthy navigation instructions, often at the expense of robust, fundamental navigation capabilities.** This is evidenced by the fact that SOTA models frequently fail to execute simple, direct instructions (see Figure 1 (a)). We have supplemented our analysis with an additional successful visual case study (see Appendix B.2 and Figure 8). This case demonstrates that the model does not strictly adhere to the guidance of instructions during the execution of long instructions. This further confirms that the model over-relies on visual and textual cues while neglecting the learning of fundamental navigation skills. Using the existing training paradigm to equip large models with general navigation capabilities is problematic.

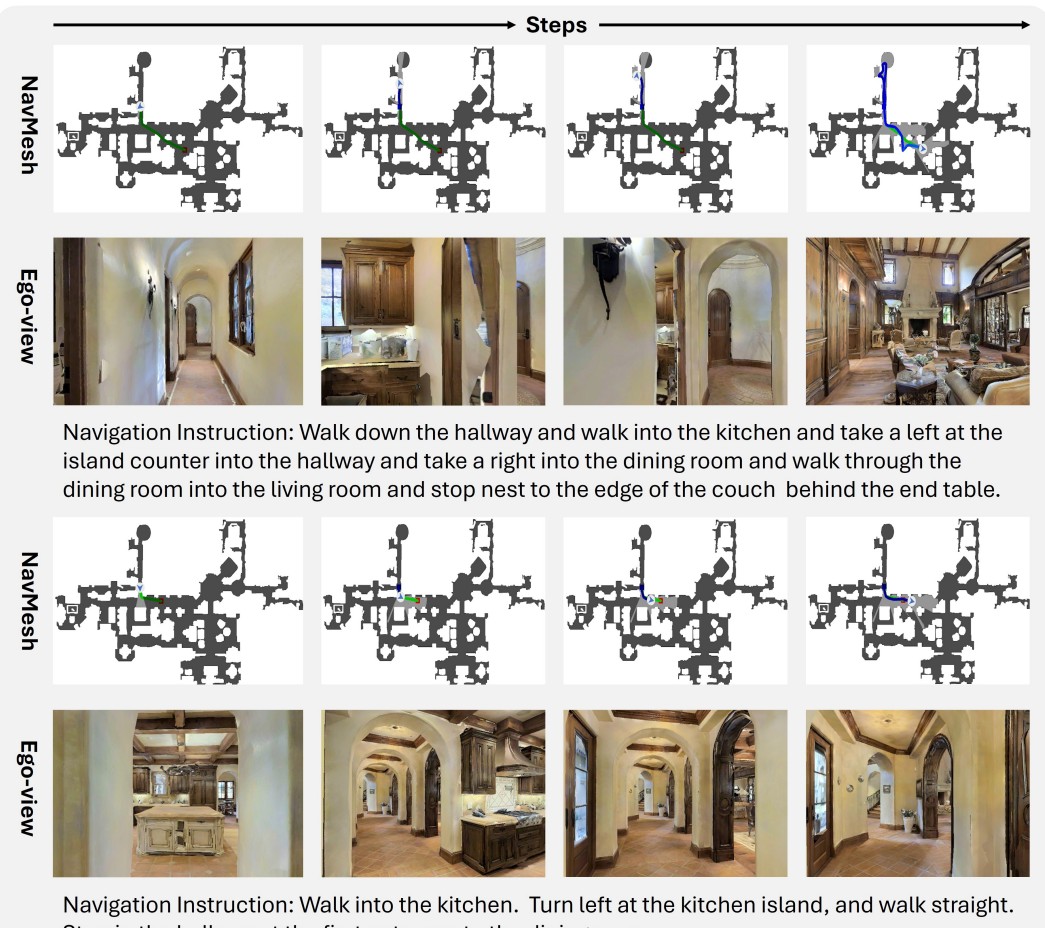

Figure 1: Visualization of successful samples in R2R. In the navigation mesh (NavMesh), the color green denotes the reference path, while blue indicates the actual path taken by the agent.

**Therefore, this work aims to refocus attention on enhancing the basic navigation abilities of end-to-end models within a MLLMs framework. We propose constructing corresponding**

**datasets and evaluation standards to ultimately foster the development of general-purpose navigation models.**

**Comparison with Alternative Approaches.** Regarding alternative existing approaches, these methods often lack a unified understanding and execution of language, typically requiring additional modules for semantic alignment. For instance, **UniGoal** (Yin et al., 2025) (accepted to CVPR 2025), presented as a zero-shot general-purpose navigation framework, achieves cross-task navigation through unified graph representations and LLM reasoning. **However, its performance is heavily dependent on the quality of dynamic scene graph construction, limiting its ability to handle semantically rich tasks. This is reflected in its evaluation, which includes only 5 object types and achieves a success rate of just 20% on text-based navigation (TextNav).** In contrast, Video-LLM-based approaches take only local observations and a navigation instruction as input to directly output actions. The VLMB dataset constructed in our work contains descriptions involving over 800 object types and spatial relations, providing a richer semantic foundation. We acknowledge the strengths of alternative models in specific tasks, which inspired our effort to build a more comprehensive Video-LLM-based VLN model. Moreover, we must emphasize that completing navigation tasks directly from instructions without relying on external positioning information or additional vision-language understanding modules is a capability where other models consistently underperform.

**Weakness 2: Supplementary Experiments and Dataset Analysis**

StreamVLN and NaVid are primarily trained on VLN-CE (R2R and RxR), and their extended datasets. We conducted a detailed analysis of instruction composition in VLN-CE. Analysis (see Appendix B.1) shows that the R-Nav component in VLMB effectively decomposes and reconstructs the instruction patterns found in VLN-CE, breaking long multi-segment instructions into shorter, executable commands. **Therefore, R-Nav (instructions collected directly without labeling, e.g., move to the desk) is essentially a subset of VLN-CE, and these instructions are already partially represented in existing model training sets. We observe that SOTA models handle short instructions less effectively than long ones (see Figure 1 (b)). This finding precisely highlights our key point: current training overemphasizes image-text matching at the expense of basic action learning.**

Furthermore, the suggestion you raised is highly valuable. Accordingly, we have supplemented the evaluation experiments of "Move-to-Anything" on the R2R dataset in Section 5.4 and Table 4. **The results demonstrate that even with minimal external data, our approach has improved performance on the R2R benchmark.**

Table 3: Comparison with SOTA methods on VLN-CE R2R Val-Unseen split.

| Method | Traning Dataset | External Data | R2R Val-Unseen | | | |
|---|---|---|---|---|---|---|
| | | | SR↑ | SPL↑ | OS↑ | NE↓ |
| NaVid (Zhang et al., 2024a) | R2R+RxR+VLNdata | 953K | 37.4 | 25.9 | 49.1 | 5.47 |
| StreamVLN (Wei et al., 2025) | R2R+RxR+EnvDrop | 10033K | 45.5 | 41.6 | 53.8 | 6.05 |
| Move-to-Anything* (ours) | R2R+RxR | 0k | 33.6 | 30.1 | 43.1 | 7.00 |
| Move-to-Anything (ours) | R2R+RxR+VLMB | 40K | 38.0 (**+4.4**) | 34.5 (**+4.4**) | 43.2 (**+0.1**) | 6.45 (**-0.55**) |

**Weakness 3: Core Problem Addressed by VLMB**

We contend that the core issue in current research may stem from a fundamental misjudgment of the primary challenges in MLLM-based VLN models. The prevailing paradigm treats the alignment between long instructions and navigation observations as the central challenge in VLN, leading researchers to concentrate on training and evaluation using long-horizon navigation data. **Our novelty lies in demonstrating that a primitive-based curriculum is a more effective path to solving the hard long-horizon problem than end-to-end training alone.**

We emphasize that while R-Nav (instructions collected directly without labeling, e.g., move to the desk) decomposes long-horizon instructions, it does not simplify the navigation task itself. **Instead, VLMB expands the diversity of goal distributions, spatial (e.g., orientation, position, region) and semantics (e.g., color, size, material) in single-step navigation, refocusing research on the execution of basic commands.** The Move-to-Anything model serves as a closed-loop validation of the identified problem. **To ensure a fair comparison, our methodological innovations are intentionally minimal yet effective, designed to introduce no additional training overhead.**

**Responses to Additional Questions**

**Q1: The classification of Modular Methods**

We have refined the description of Related Work (Section 2) as suggested. Furthermore, although advancements in end-to-end network architectures have led to works (Chen et al., 2021; 2022b) that build navigation systems based on deep neural networks or Transformers, these methods, in practice, still depend on external positioning systems (e.g., GPS or SLAM) or multi-sensor inputs (e.g., depth, panoramic images). **Therefore, unifying perception and planning is crucial for enhancing decision consistency and generalization, which is vital for the practical deployment of VLN.**

**Q2: Scene Selection Criteria**

VLN tasks are inherently designed for semantically rich indoor environments, and MP3D and HM3D are widely adopted benchmarks specifically built for such settings. To maximize the utility of these resources, we have extended the semantic (e.g., color, size, material) and spatial (e.g., orientation, position, region) descriptions within these scenes by building an automated collection and annotation process. **Compared to existing datasets, our constructed dataset offers significantly richer object categories and more detailed descriptions (as shown in Table 1 and Figure 3), thereby enhancing diversity and expressiveness while maintaining compatibility with standard VLN environments.**

**Q3: Standardizing the Use of the Term ObjNav**

Following the suggestion, we have renamed the initial navigation instructions (e.g., *move to the desk*) to Raw Navigation (R-Nav) instructions.

**Q4: Statistical Analysis of Common Instruction Formats**

We conducted a quantitative analysis of common instruction formats based on R2R and RxR, along with a detailed description of the definitions and classification methodology for the four instruction types, which is included in Appendix B.1 and Table 7.

Table 4: Proportions of common navigation primitive types.

| Navigation Primitive | Instruction Numbers | Percentage |
|---|---|---|
| Move | 227,189 | 42% |
| Situation | 200,143 | 37% |
| Change Direction | 70,320 | 13% |
| Change Region | 43,274 | 8% |

**Q5: Using Navigation Primitives to Solve Object-centric Tasks**

We conducted a thorough analysis of common instruction instruction composition (summarized in Appendix B.1), confirming that the "move-to" primitive covers most navigation scenarios. For object-centric tasks like REVERIE (a more challenging navigation and localization task), we believe that equipping models with fundamental navigation skills and leveraging LLM reasoning could address this within our proposed framework. We sincerely appreciate this suggestion and will prioritize REVERIE in future evaluations. However, we wish to emphasize that basic, in-field-of-view navigation remains an overlooked aspect in Video-LLM training. Our work is the first to identify this issue and provide a demonstrated solution.

We are grateful for the reviewer's valuable feedback, which has significantly strengthened our manuscript.

## Response to Reviewer 3:

We greatly appreciate the reviewer's thoughtful comments and valuable suggestions. The manuscript has been thoroughly revised to address all raised concerns. Detailed point-by-point responses are provided below, and all corresponding changes in the main text are highlighted in **Green**.

**Weakness 1: Impact of Instruction Length on VLA Model Performance**

We conducted an instruction decomposition analysis on VLN-CE (R2R and RxR), as shown in Appendix B.1. The analysis reveals that while these datasets exhibit linguistic diversity, the instructions suffer from vague descriptions, ill-defined sub-goals, and significant homogeneity in target categories. In fact, since VLN-CE was designed long before the maturity of MLLMs, it was not constructed following the training paradigm widely recognized in VLA research—where models first acquire basic skills (e.g., pick up, put down) through foundational action training before progressing to multi-step reasoning for long-horizon tasks. Moreover, due to instructional ambiguity, training tends to overfit to visual observations and navigation instructions, resulting in many "successful" trajectories that reach the final goal without accomplishing intermediate sub-goals. **Consequently, current evaluation paradigms cannot accurately assess a model's success rate in achieving individual navigation sub-goals, as shown in Figure 8.**

To address these issues, we prioritize rectifying the training paradigm. By constructing "move-to" navigation instructions, we aim to equip the model with stable goal-oriented navigation capabilities based on single-step directives. These segmented "move-to" instructions are then combined to accomplish long-horizon tasks. As shown in Table 3, when consecutive sub-goals increase (e.g., three sub-goals), the overall success rate naturally declines. **However, we emphasize that in practical navigation, future sub-goals are updated incrementally based on ongoing environmental observations. With new instructions issued dynamically in response to navigation observations, the success rate remains consistent.** For this reason, we deliberately restrict the training set to single-step navigation instructions and introduce multi-goal instructions only during evaluation. This approach ensures that executing sequential single-step navigation tasks can fulfill long-term demands while avoiding overfitting to image-instruction alignment—where models might bypass intermediate targets and proceed directly to the final goal.

**Weakness 2: The Concept of Using Short-term and Long-term Frames**

To balance recent detail fidelity with global context in long-horizon reasoning, we adopt a hierarchical memory that explicitly separates short-term (ST) and long-term (LT) observations. Concretely, the most recent short-term window ($W$) frames are encoded by a frozen vision tower and retained as high-fidelity multi-token visual embeddings, which are injected into the LLM (as `<image>`) to remain sensitive to immediate environmental changes. **Earlier frames are not stored in full. Instead, each frame is first globally pooled into a per-frame feature. These features are then aggregated via a lightweight mean pooling and an MLP into a fixed number of long-term memory slots ($M$). This process, summarized as `<history>`, maintains essential global context at a low token cost.** Consequently, the memory does *not* keep all tokens from all frames: only the $W$ most recent frames preserve multi-token detail, while prior frames are compressed into $M$ slots. This decoupling reduces token complexity from $O(T \times S)$ to $O(W \times S + M)$, where $T$ is the total number of frames and $S$ is the spatial token count, substantially lowering computational and memory costs without sacrificing useful recent detail.

We have supplemented ablation experiments to clarify this issue. Specifically, Table 6 reports results when varying the short-term window size ($W$) and the number of long-term memory slots ($M$). We also compare our default mean+MLP aggregation strategy with a no-aggregation baseline. The findings show that increasing $W$ and $M$ improves navigation performance up to an optimal configuration ($W$=8, $M$=1568). Moreover, the mean+MLP aggregator achieves a favorable balance between preserving hierarchical memory and maintaining computational efficiency.

Table 5: Ablation study on HM configuration: impact of short-term window size ($W$), long-term memory slots ($M$), and aggregation method.

| | | | VLMB | | | | | | | |
|---|---|---|---|---|---|---|---|---|---|---|
| | | | MP3D | | | | HM3D | | | |
| $W$ | $M$ | Aggregator Type | SR↑ | SPL↑ | OS↑ | NE↓ | SR↑ | SPL↑ | OS↑ | NE↓ |
| 4 | 784 | | 53.2 | 51.8 | 60.3 | 3.78 | 64.7 | 63.1 | 69.2 | 3.12 |
| 8 | 784 | Mean+MLP | 57.1 | 55.7 | 64.4 | 3.45 | 68.9 | 67.5 | 73.4 | 2.89 |
| 8 | 1,568 | | **60.6** | **59.6** | **67.6** | **3.21** | **71.4** | **70.3** | **76.3** | **2.71** |
| 16 | 1,568 | | 58.8 | 57.9 | 65.9 | 3.38 | 70.6 | 69.4 | 75.1 | 2.74 |
| 8 | 1,568 | N/A | 57.7 | 56.5 | 64.2 | 3.40 | 68.9 | 67.6 | 73.8 | 3.01 |

**Weakness 3: Dataset Built on Off-the-shelf Scenes**

We acknowledge that our dataset is collected using off-the-shelf scenes and simulators. However, as you noted, VLN tasks are inherently designed for semantically rich indoor environments, and MP3D and HM3D are widely adopted benchmarks specifically built for such settings. **To maximize the utility of these resources, we have extended the semantic (e.g., color, size, material) and spatial (e.g., orientation, position, region) descriptions within these scenes by building an automated collection and annotation process.** Compared to existing datasets, our constructed dataset offers significantly richer object categories and more detailed descriptions (as shown in Table1 and Figure3), thereby enhancing diversity and expressiveness while maintaining compatibility with standard VLN environments.

**Responses to Additional Questions**

**Q1: Supplementary Experiments and Dataset Analysis**

We have verified that the compact dataset proposed in this paper effectively enhances the model's fundamental navigation skills. To further validate the contribution of the VLMB dataset to navigation capability, we conducted joint training using VLMB together with R2R and RxR datasets—even though we maintain that the instruction composition of R2R alone is insufficient for equipping models with general navigation competence. **Experimental results (Table 4) show that by incorporating even a small amount of VLMB data during joint training, the model achieves higher performance levels on R2R, demonstrating the usefulness and effectiveness of the VLMB dataset.**

- A model (LLaVA-Vid) trained on R2R+RxR evaluated on R2R
- A model (LLaVA-Vid) trained on R2R+RxR+VLMB evaluated on R2R

Table 6: Comparison with SOTA methods on VLN-CE R2R Val-Unseen split.

| Method | Traning Dataset | External Data | R2R Val-Unseen | | | |
|---|---|---|---|---|---|---|
| | | | SR↑ | SPL↑ | OS↑ | NE↓ |
| NaVid (Zhang et al., 2024a) | R2R+RxR+VLNdata | 953K | 37.4 | 25.9 | 49.1 | 5.47 |
| StreamVLN (Wei et al., 2025) | R2R+RxR+EnvDrop | 10033K | 45.5 | 41.6 | 53.8 | 6.05 |
| Move-to-Anything* (ours) | R2R+RxR | 0k | 33.6 | 30.1 | 43.1 | 7.00 |
| Move-to-Anything (ours) | R2R+RxR+VLMB | 40K | 38.0 (**+4.4**) | 34.5 (**+4.4**) | 43.2 (**+0.1**) | 6.45 (**-0.55**) |

**Q2: Analysis of Training Dataset**

StreamVLN and NaVid are primarily trained on VLN-CE (R2R and RxR), and their extended datasets. We conducted a detailed analysis of instruction composition in VLN-CE, and this analysis (see Appendix B.1) shows that the R-Nav component in VLMB effectively decomposes and reconstructs the instruction patterns found in VLN-CE, breaking long multi-segment instructions into shorter, executable commands. **Therefore, R-Nav (instructions collected directly without labeling, e.g., *move to the desk*) is essentially a subset of VLN-CE, and these instructions are already included in existing model training sets.** However, while current SOTA models perform well on long-instruction tasks, they struggle on this subset—precisely highlighting our key argument: existing training paradigms overemphasize image-text alignment while neglecting the learning of fundamental navigation actions.

The constructive feedback provided by the reviewer has greatly improved the rigor and clarity of our work, for which we are deeply thankful.

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

Navigation Instruction: Walk down the hallway and walk into the kitchen and take a left at the island counter into the hallway and take a right into the dining room and walk through the dining room into the living room and stop nest to the edge of the couch behind the end table.

Navigation Instruction: Walk into the kitchen. Turn left at the kitchen island, and walk straight. Stop in the hallway at the first entrance to the dining room.

Figure 8: Visualization of successful samples in R2R. In the navigation mesh (NavMesh), the color green denotes the reference path, while blue indicates the actual path taken by the agent.

## C  VISUALIZATION RESULTS

We selected a subset of samples from the HM3D scenes for visualization, as shown in Figure 9. These samples include various spatial and semantic characteristics: directional relations (e.g., *on the left side of the room*), regional cues (e.g., *visible through the doorway straight ahead*), objects located at the edge of the observation field (often easily overlooked), and region containment relations (e.g., *in the kitchen*). Together, these examples represent a broad range of spatial and semantic information commonly encountered in navigation tasks.

We further visualize samples from multi-step instruction navigation tasks, as shown in Figure 10. In some cases, a noticeable gap exists between the actual navigation path taken by the agent and the optimal path generated by the simulator. This observation suggests that the model is not merely memorizing predefined routes but instead demonstrates a certain level of path planning capability. These results support the effectiveness of the navigation primitive-based paradigm in building general-purpose navigation systems.

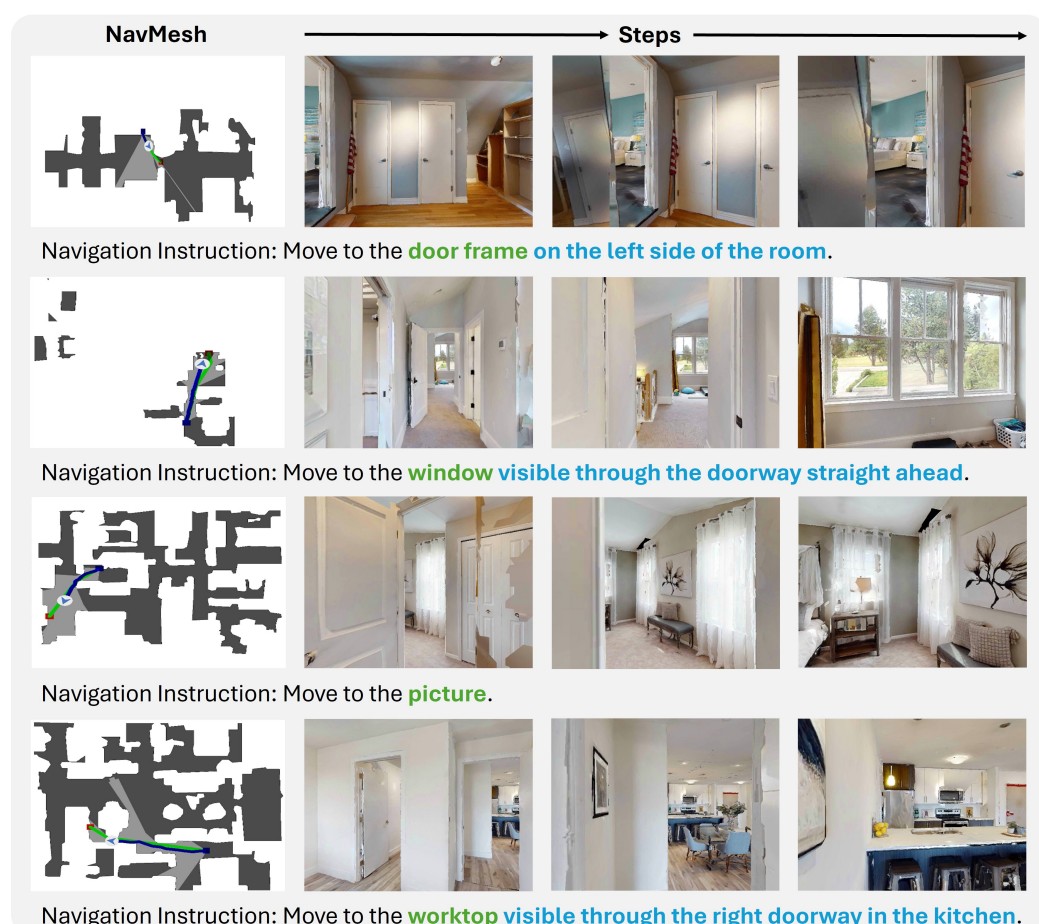

Figure 9: Examples of visualized trajectories and observed images in HM3D.

## D  USE OF LARGE LANGUAGE MODELS (LLMS)

We used Large Language Models (LLMs) in two limited ways: (1) ChatGPT was employed to polish the language and improve clarity of the manuscript; (2) During dataset construction, ChatGPT assisted in generating candidate instructions and validating specific annotation steps under human supervision. All outputs were manually reviewed to ensure accuracy and integrity. No experimental results or analyses were produced solely by LLMs.

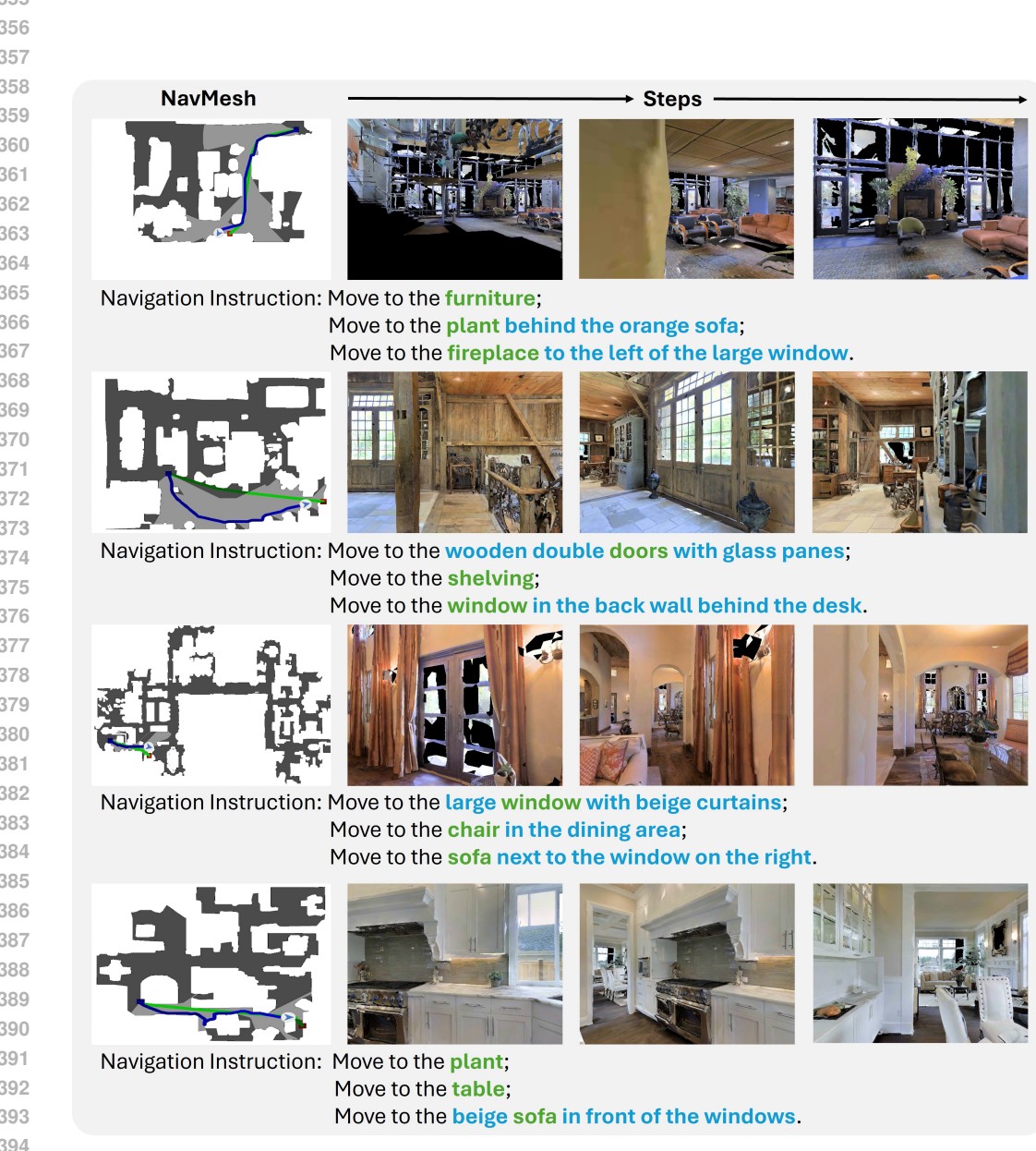

Figure 10: Examples of visualized trajectories and observed images in a multi-step navigation task.