# OpenReview forum: "From End-to-End to Step-by-Step: learning Composable Navigation Primitives for Vision-Language Navigation"
_ICLR.cc/2026/Conference — Submitted to ICLR 2026_

### Official Review · Reviewer_NPKL · 2025-10-27

**Soundness:** 3
**Presentation:** 3
**Contribution:** 3
**Rating:** 6
**Confidence:** 4

**Summary:**

This paper identifies a phenomenon in current video-based navigation VLAs: although they perform well in long-horizon navigation, they exhibit poor performance in short-horizon navigation tasks. The authors believe the reason lies in the unbalanced training data of VLN-CE R2R and RxR. To address this, they propose the VLMB dataset, which incorporates diverse instruction horizons across both MP3D and HM3D environments. Moreover, the authors propose a video-based VLA that efficiently models both short-term and long-term history, achieving best performance on the VLMB benchmark.

**Strengths:**

- The highlighted phenomenon is interesting and significant in current VLN-oriented VLA methods, as it reflects the data foundations underlying existing approaches.
- The proposed dataset, VLMB, could benefit the research community by providing more diverse instructions and scenarios.
- The Move-to-Anything framework is largely technically sound and demonstrates better performance than existing methods.

**Weaknesses:**

Major weakness:
- The paper stops at comparing short-horizon and long-horizon tasks. I would like to see a more in-depth analysis of the impact of instruction length (i.e., the number of sub-goals) on VLA models—such as the upper limit of sub-goal size that existing VLA models can handle, or the performance variance across different numbers of sub-goals.
- The concept of using short-term and long-term frames is unclear to me. How are long-term observations maintained (e.g., are all frames kept, or is there a selection process)? Does the memory retain all tokens, or only those from chosen frames?

Minor weakness:
- The trajectories and instructions are based on off-the-shelf scenes and simulators, which diminishes the dataset's contribution. However, I believe this is acceptable since the proposed data are tailored for a specific purpose.

**Questions:**

- All experiments are conducted in the VLMB environment, which raises the question of how the proposed method would perform on VLN-CE benchmarks such as R2R or RxR. Would training on VLMB data improve model performance in VLN-CE, for instance, through co-training with data from both datasets?
- The comparison appears unfair, as StreamVLN and NaVid have not been fine-tuned on VLMB.

---

> ### Author Response · Authors · 2025-11-20
> **Response to Reviewer NPKL (Part1/2)**
>
> We sincerely thank the reviewer for their thoughtful comments and valuable suggestions. We have provided point-by-point responses below. We are happy to discuss any of these points further. We have also updated the manuscript PDF to address the raised concerns. The revised sections are highlighted in green in the supplementary PDF file.
>
> >**Weakness1: Impact of Instruction Length on VLA Model Performance**
>
> - We analyzed the instructions in VLN-CE (R2R and RxR), as shown in **Appendix B.1**. This analysis shows that while the datasets have linguistic diversity, their instructions often contain vague descriptions and ill-defined sub-goals. The target categories also show significant homogeneity. VLN-CE was created before multimodal LLMs became mature. Therefore, its design does not follow the VLA training paradigm, where models first learn basic skills through foundational action training (e.g., pick up, put down). They then progress to multi-step reasoning for long tasks. Furthermore, ambiguous instructions cause overfitting to visual and navigation cues. This leads to trajectories that **reach the final goal without completing intermediate sub-goals**. Thus, current evaluation paradigms cannot accurately measure a model's success in achieving individual sub-goals, as seen in **Figure 8**.
>
> - To solve these problems, we focus on correcting the training approach. First, we build "move-to" navigation instructions. This gives the model stable goal-oriented navigation skills based on single-step commands. These single-step instructions are then combined to perform long-horizon tasks. **Table 3**​ shows that the overall success rate drops as more sub-goals are added (e.g., three sub-goals). However, in real navigation, **later sub-goals are updated based on new environmental observations**. When instructions are given dynamically during navigation, the success rate stays consistent.​ Therefore, we use only **single-step instructions for training**. **Multi-goal instructions are used only for evaluation**.​ This method ensures that performing a sequence of single-step tasks can meet long-term needs. It also prevents overfitting to the image-instruction alignment. Without this, models might skip intermediate targets and go straight to the final goal.
>
> >**Weakness2: The Concept of Using Short-term and Long-term Frames**
>
> - To balance recent detail fidelity with global context in long-horizon reasoning, we adopt a hierarchical memory that explicitly separates short-term (ST) and long-term (LT) observations. Concretely, the most recent short-term window ($W$) frames are encoded by a frozen vision tower and retained as high-fidelity multi-token visual embeddings, which are injected into the LLM (as \<image>) to remain sensitive to immediate environmental changes. **Earlier frames are not stored in full**. Instead, each frame is first **globally pooled into a per-frame feature**. These features are then aggregated via a lightweight mean pooling and an MLP into a fixed number of long-term memory slots ($M$). This process, summarized as \<history>, maintains essential global context at a low token cost.} Consequently, the memory does \emph{not} keep all tokens from all frames: only the $W$ most recent frames preserve multi-token detail, while prior frames are compressed into $M$ slots. This decoupling reduces token complexity from $O(T \times S)$ to $O(W \times S + M)$, where $T$ is the total number of frames and $S$ is the spatial token count, substantially lowering computational and memory costs without sacrificing useful recent detail.
>
> - We have supplemented ablation experiments to clarify this issue. Specifically, **Table 6** reports results when varying the short-term window size ($W$) and the number of long-term memory slots ($M$). We also compare our default mean+MLP aggregation strategy with a no-aggregation baseline. The findings show that increasing $W$ and $M$ improves navigation performance up to an optimal configuration ($W{=}8$, $M{=}1568$). Moreover, the mean+MLP aggregator achieves a favorable balance between preserving hierarchical memory and maintaining computational efficiency.
>  | W  | M    | Aggregator Type | MP3D | | | | | HM3D | | | |
> |----|------|-----------------|------|-|-|-|-|------|-|-|-|
> |    |      |                 | SR↑ | SPL↑ | OS↑ | NE↓ | | SR↑ | SPL↑ | OS↑ | NE↓ |
> |----|------|-----------------|-----|------|-----|-----|-|-----|------|-----|-----|
> | 4  | 784  | Mean+MLP        | 53.2 | 51.8 | 60.3 | 3.78 | | 64.7 | 63.1 | 69.2 | 3.12 |
> | 8  | 784  | Mean+MLP        | 57.1 | 55.7 | 64.4 | 3.45 | | 68.9 | 67.5 | 73.4 | 2.89 |
> | 8  | 1,568| Mean+MLP        | **60.6** | **59.6** | **67.6** | **3.21** | | **71.4** | **70.3** | **76.3** | **2.71** |
> | 16 | 1,568| Mean+MLP        | 58.8 | 57.9 | 65.9 | 3.38 | | 70.6 | 69.4 | 75.1 | 2.74 |
> | 8  | 1,568| N/A             | 57.7 | 56.5 | 64.2 | 3.40 | | 68.9 | 67.6 | 73.8 | 3.01 |

---

> ### Author Response · Authors · 2025-11-20
> **Response to Reviewer NPKL (Part2/2)**
>
> >**Weakness3: Dataset Built on Off-the-shelf Scenes**
>
> - We acknowledge that our dataset is collected using off-the-shelf scenes and simulators. However, as you noted, VLN tasks are inherently **designed for semantically rich indoor environments**, and MP3D and HM3D are widely adopted benchmarks specifically built for such settings. To maximize the utility of these resources, we have extended the **semantic (e.g., color, size, material) and spatial (e.g., orientation, position, region) descriptions** within these scenes by building an automated collection and annotation process. Compared to existing datasets, our constructed dataset offers significantly richer object categories and more detailed descriptions (as shown in **Table1**  and **Figure 3**), thereby enhancing diversity and expressiveness while maintaining compatibility with standard VLN environments.
>
> ### **Responses to Additional Questions**
> >**Q1: Supplementary Experiments and Dataset Analysis**
>
> - We have verified that the compact dataset proposed in this paper effectively enhances the model's fundamental navigation skills. To further validate the contribution of the VLMB dataset to navigation capability, we conducted joint training using VLMB together with R2R and RxR datasets—even though we maintain that the instruction composition of R2R alone is insufficient for equipping models with general navigation competence. Experimental results (**Table 4**) show that by incorporating even **a small amount of VLMB data during joint training**, the model achieves **higher performance levels on R2R, demonstrating the usefulness and effectiveness of the VLMB dataset**.
> | Method                      | Training Dataset     | External Data | SR↑               | SPL↑              | OS↑               | NE↓               |
> | :-------------------------- | :------------------- | :------------ | :---------------- | :---------------- | :---------------- | :---------------- |
> | NaVid                       | R2R+RxR+VLNdata      | 953K          | 37.4              | 25.9              | 49.1              | 5.47              |
> | StreamVLN                   | R2R+RxR+EnvDrop      | 10033K        | 45.5              | 41.6              | 53.8              | 6.05              |
> | Move-to-Anything* (ours)    | R2R+RxR              | 0k            | 33.6              | 30.1              | 43.1              | 7.00              |
> | Move-to-Anything (ours)     | R2R+RxR+VLMB         | 40K           | 38.0 (**+4.4**)   | 34.5 (**+4.4**)   | 43.2 (**+0.1**)   | 6.45 (**-0.55**)  |
>
> >**Q2: Analysis of Training Dataset**
>
> - StreamVLN and NaVid are primarily trained on VLN-CE (R2R and RxR), and their extended datasets. We conducted a detailed analysis of instruction composition in VLN-CE, and this analysis (see **Appendix B.1**) shows that the R-Nav component in VLMB effectively decomposes and reconstructs the instruction patterns found in VLN-CE, breaking long multi-segment instructions into shorter, executable commands. Therefore, R-Nav (instructions collected directly without labeling, e.g., move to the desk) is essentially **a subset of VLN-CE, and these instructions are already included in existing model training sets**. Current SOTA models perform well on long-instruction tasks, yet they struggle on this subset. This performance gap supports our key argument. We contend that existing training paradigms show an **overemphasis on image-text alignment, at the expense of learning fundamental navigation actions**.
>
> The constructive feedback provided by the reviewer has greatly improved the rigor and clarity of our work, for which we are deeply thankful.

---

> > ### Comment · Reviewer_NPKL · 2025-11-27
> > **Response to authors**
> >
> > Thanks for the thorough response and my concerns have been addressed.
> >
> > Overall, this paper investigates an interesting problem in VLN. The proposed framework works, and the data could help the community better evaluate methods in this field.
> >
> > I maintain my positive score.

---

> ### Author Response · Authors · 2025-11-27
>
> We sincerely appreciate your recognition of our work. We will further improve the paper based on your valuable comments. Thank you again.

---

### Official Review · Reviewer_XQBp · 2025-10-27

**Soundness:** 3
**Presentation:** 3
**Contribution:** 2
**Rating:** 4
**Confidence:** 5

**Summary:**

This paper introduces a new framework called Move-to-Anything and a corresponding dataset named VLMB, aimed at improving Vision-and-Language Navigation by enabling models to execute general, step-by-step instructions in a more flexible and human-like manner. The authors argue that traditional VLN methods struggle with simple, generalizable commands due to their dependency on fixed instruction distributions. To address this, they reformulate VLN into a more modular and universal navigation task, where agents are trained to move toward any visual or linguistic target. The proposed method integrates spatial grounding and multimodal understanding, and the experiments show consistent performance gains over baseline models.

**Strengths:**

1. Clear Presentation and Visualization. The paper is clearly written and well-organized. The figures and examples effectively illustrate the motivation, architecture, and qualitative results. The visualizations, especially those comparing instruction-following behaviors, help readers intuitively understand the model’s capabilities.
2. Improved Performance in Experiments. The proposed Move-to-Anything demonstrates noticeable performance improvements over strong  VLN baselines across the proposed dataset. The results validate that the method is better at handling flexible, goal-driven instructions.
3. Reframing VLN to a generalized primitive-based paradigm is an interesting idea. It pushes the direction of VLN research toward more open-ended and human-like reasoning rather than narrow, dataset-dependent training.

**Weaknesses:**

1. Motivation. The biggest issue with this paper is the motivation. The authors claim that “existing models fail to execute these general but straightforward instructions effectively,” but this statement is not well supported by experiments. The results in Table 2 only compare two VLN methods—NaVid and StreamVLN—while many other existing approaches are not included. Without broader evidence, it’s hard to accept the claim that current models generally fail on such instructions.
2. Dataset Distribution and Fairness. VLN models are known to be highly sensitive to the distribution of instructions, which is why we usually need to train a separate model for each dataset. For example, a model trained on R2R performs poorly on RxR, and I think this is quite similar to the situation described in the paper. The VLMB dataset seems to be another dataset with a different instruction distribution. So, it’s not surprising that the Move-to-Anything model, when trained on VLMB, performs much better than models that were never exposed to this type of data. I’m also curious how the proposed method would perform on R2R or RxR, which are also multi-step navigation tasks (as in Table 3).
3. Novelty. The novelty of the paper is limited. The main challenge in VLN is to align language instructions with visual observations over long trajectories, while the proposed dataset seems to simplify the task instead of addressing this difficulty. In addition, the two new components added to the Move-to-Anything model are techniques already widely used in MLLMs and VLN research, so the technical contribution feels incremental.

**Questions:**

1. Classification of Modular Methods. In the “Related Work” section, the term Modular Method VLN itself feels a bit odd. The paper describes these methods as ones that “decompose the navigation pipeline into distinct components such as perception, language understanding, mapping, planning, and control.” However, the cited examples—HAMT and DUET—are generally regarded as end-to-end approaches. I think the criteria for this classification need to be explained more clearly.
2. Scene Selection Criteria. The paper does not mention the standard for selecting scenes from MP3D and HM3D.
3. The authors should avoid using the term ObjNav to describe one type of generated instruction. It’s confusing because ObjNav is also used in the paper with a different meaning.
4. The finding in lines 92-93 is not mentioned and explained in the latter part.
5. Currently, the primitives are limited to "MoveTo", is there any other possible primitives or how could we use such primitives to handle object-centric tasks like REVERIE?

---

> ### Author Response · Authors · 2025-11-20
> **Response to Reviewer XQBp (Part1/3)**
>
> We sincerely thank the reviewer for their insightful comments and constructive feedback. We have carefully revised the manuscript to address the concerns raised. Any further discussion will be appreciated. Besides, we have updated the PDF of our paper to address your concerns. The corresponding modified sections in the manuscript (see PDF in Supplementary Material) have been marked in Red.
>
> > **Weakness1: Clarification of Motivation and Comparison with Alternative Approaches**
>
> - **Clarification of Motivation.** We acknowledge that the initial motivation presented in the paper may have been insufficiently elaborated. Our core argument is that the current training paradigm for end-to-end VLN models, particularly those based on Video-LLM-based frameworks, is predominantly evaluated on similar datasets (e.g., R2R, RxR). This narrow focus has led models to **overemphasize the alignment between visual inputs and lengthy navigation instructions, often at the expense of robust, fundamental navigation capabilities**. This is evidenced by the fact that SOTA models frequently fail to execute simple, direct instructions (see **Figure1 (a)**). We have supplemented our analysis with an additional successful visual case study (see **Appendix B.2** and **Figure 8**). This case demonstrates that the model does not strictly adhere to the guidance of instructions during the execution of long instructions. This further confirms that the model over-relies on visual and textual cues while neglecting the learning of fundamental navigation skills. Using the existing training paradigm to equip large models with general navigation capabilities is problematic.
> Therefore, this work aims to **refocus attention on enhancing the basic navigation abilities of end-to-end models within a MLLMs framework**. We propose constructing corresponding datasets and evaluation standards to ultimately foster the development of general-purpose navigation models.
>
> - **Comparison with Alternative Approaches.** Regarding alternative existing approaches, these methods often lack a unified understanding and execution of language, typically requiring additional modules for semantic alignment. For instance, **UniGoal** [1] (accepted to CVPR 2025), presented as a zero-shot general-purpose navigation framework, achieves cross-task navigation through unified graph representations and LLM reasoning. However, its performance is heavily dependent on the quality of dynamic scene graph construction, limiting its ability to handle semantically rich tasks. This is reflected in its evaluation, which **includes only 5 object types and achieves a success rate of just 20\% on text-based navigation (TextNav)**. In contrast, Video-LLM-based approaches take only local observations and a navigation instruction as input to directly output actions. The VLMB dataset constructed in our work contains descriptions involving over 800 object types and spatial relations, providing a richer semantic foundation. We acknowledge the strengths of alternative models in specific tasks, which inspired our effort to build a more comprehensive Video-LLM-based VLN model. Moreover, we must emphasize that completing navigation tasks directly from instructions without relying on external positioning information or additional vision-language understanding modules is a capability where other models consistently underperform.

---

> ### Author Response · Authors · 2025-11-20
> **Response to Reviewer XQBp (Part2/3)**
>
> > **Weakness2: Supplementary Experiments and Dataset Analysis**
> - StreamVLN [2] and NaVid [3] are primarily trained on VLN-CE (R2R and RxR), and their extended datasets. We conducted a detailed analysis of instruction composition in VLN-CE. Analysis (see **Appendix B.1**) shows that the R-Nav component in VLMB effectively decomposes and reconstructs the instruction patterns found in VLN-CE, breaking long multi-segment instructions into shorter, executable commands. Therefore, **R-Nav (instructions collected directly without labeling, e.g., move to the desk) is essentially a subset of VLN-CE**, and these instructions are already partially represented in existing model training sets. We observe that SOTA models handle short instructions less effectively than long ones (see **Figure1 (b)**). This finding precisely highlights our key point: current training overemphasizes image-text matching at the expense of basic action learning.
> - Furthermore, the suggestion you raised is highly valuable. Accordingly, we have supplemented the evaluation experiments of “Move-to-Anything" on the R2R dataset in **Section 5.4** and **Table 4**. The results demonstrate that even with minimal external data, our approach has **improved performance on the R2R benchmark**.
> | Method                      | Training Dataset     | External Data | SR↑               | SPL↑              | OS↑               | NE↓               |
> | :-------------------------- | :------------------- | :------------ | :---------------- | :---------------- | :---------------- | :---------------- |
> | NaVid                       | R2R+RxR+VLNdata      | 953K          | 37.4              | 25.9              | 49.1              | 5.47              |
> | StreamVLN                   | R2R+RxR+EnvDrop      | 10033K        | 45.5              | 41.6              | 53.8              | 6.05              |
> | Move-to-Anything* (ours)    | R2R+RxR              | 0k            | 33.6              | 30.1              | 43.1              | 7.00              |
> | Move-to-Anything (ours)     | R2R+RxR+VLMB         | 40K           | 38.0 (**+4.4**)   | 34.5 (**+4.4**)   | 43.2 (**+0.1**)   | 6.45 (**-0.55**)  |
>
> >**Weakness3: Core Problem Addressed by VLMB**
> - We contend that the core issue in current research may stem from a fundamental misjudgment of the primary challenges in MLLM-based VLN models. The prevailing paradigm treats the alignment between long instructions and navigation observations as the central challenge in VLN, leading researchers to concentrate on training and evaluation using long-horizon navigation data.
> Our novelty lies in **demonstrating that a primitive-based curriculum is a more effective path to solving the hard long-horizon problem than end-to-end training alone**.
> - We emphasize that while R-Nav (instructions collected directly without labeling, e.g., move to the desk) decomposes long-horizon instructions, it does not **simplify the navigation task** itself. Instead, VLMB expands the diversity of goal distributions, spatial (e.g., orientation, position, region) and semantics (e.g., color, size, material) in single-step navigation, refocusing research on the execution of basic commands. Meanwhile, by combining single-step move-to operations, we can automatically generate navigation instructions with **varying steps and cycle lengths**, thereby enhancing the model's training and evaluation processes. The Move-to-Anything model serves as a closed-loop validation of the identified problem. To ensure a fair comparison, our methodological innovations are **intentionally minimal yet effective**, designed to introduce no additional training overhead.

---

> ### Author Response · Authors · 2025-11-20
> **Response to Reviewer XQBp (Part3/3)**
>
> ### **Responses to Additional Questions**
>
> >**Q1: The classification of Modular Methods**
>
> - We have refined the description of Related Work (**Section 2**) as suggested.
> Furthermore, although advancements in end-to-end network architectures have led to works (hamt [4], duet[5]) that build navigation systems based on deep neural networks or Transformers, these methods, in practice, still depend on external positioning systems (e.g., GPS or SLAM) or multi-sensor inputs (e.g., depth, panoramic images). Therefore, **unifying perception and planning is crucial for enhancing decision consistency and generalization**, which is vital for the practical deployment of VLN.
>
> >**Q2: Scene Selection Criteria**
>
> - VLN tasks are inherently designed for semantically rich indoor environments, and MP3D and HM3D are widely adopted benchmarks specifically built for such settings. To maximize the utility of these resources, we have extended the semantic (e.g., color, size, material) and spatial (e.g., orientation, position, region) descriptions within these scenes by building an automated collection and annotation process. Compared to existing datasets, our constructed dataset offers significantly **richer object categories and more detailed descriptions** (as shown in **Table 1** and **Figure 3**), thereby enhancing diversity and expressiveness while maintaining compatibility with standard VLN environments.
>
> >**Q3: Standardizing the Use of the Term ObjNav**
>
> - Following the suggestion, we have renamed the initial navigation instructions to **Raw Navigation (R-Nav)** instructions.
>
> >**Q4: Statistical Analysis of Common Instruction Formats**
>
> - We conducted a quantitative analysis of common instruction formats based on R2R and RxR, along with a detailed description of the definitions and classification methodology for the four instruction types, which is included in **Appendix B.1** and **Table 7**.
> | Navigation Primitive | Instruction Numbers | Percentage |
> | :------------------: | :-----------------: | :--------: |
> | Move                 | 227,189          | 42%        |
> | Situation            | 200,143             | 37%        |
> | Change Direction     | 70,320              | 13%        |
> | Change Region        | 43,274              | 8%         |
>
> >**Q5: Using Navigation Primitives to Solve Object-centric Tasks**
>
> - We conducted a thorough analysis of common instruction instruction composition (summarized in **Appendix B.1**), confirming that the “move-to'' primitive **covers most navigation scenarios**. For object-centric tasks like REVERIE [6] (a more challenging navigation and localization task), we believe that equipping models with fundamental navigation skills and leveraging LLM reasoning could address this within our proposed framework. We sincerely appreciate this suggestion and will prioritize REVERIE in future evaluations. However, we wish to emphasize that basic, **in-field-of-view navigation** remains an overlooked aspect in Video-LLM training. Our work is the first to identify this issue and provide a demonstrated solution.
>
> We are grateful for the reviewer's valuable feedback, which has significantly strengthened our manuscript.
>
> **Reference**
>
> [1] Yin, Hang, et al. "Unigoal: Towards universal zero-shot goal-oriented navigation." Proceedings of the Computer Vision and Pattern Recognition Conference. 2025.
>
> [2] Zhang, Jiazhao, et al. "Navid: Video-based vlm plans the next step for vision-and-language navigation." Proceedings of Robotics: Science and Systems. 2024.
>
> [3] Wei, Meng, et al. "Streamvln: Streaming vision-and-language navigation via slowfast context modeling." arXiv preprint arXiv:2507.05240 (2025).
>
> [4] Chen, Shizhe, et al. "History aware multimodal transformer for vision-and-language navigation." Advances in neural information processing systems 34 (2021): 5834-5847.
>
> [5] Chen, Shizhe, et al. "Think global, act local: Dual-scale graph transformer for vision-and-language navigation." Proceedings of the IEEE/CVF Conference on Computer Vision and Pattern Recognition. 2022.
>
> [6] Qi, Yuankai, et al. "Reverie: Remote embodied visual referring expression in real indoor environments." Proceedings of the IEEE/CVF Conference on Computer Vision and Pattern Recognition. 2020.

---

> > ### Comment · Reviewer_XQBp · 2025-11-24
> > **Response to Authors**
> >
> > I appreciate the authors' effort to address the concerns. However, the core issue of data distribution and fairness remains the central weakness, and the supplemental results do not solve this concern.
> >
> > First, I think the current motivation is still based on unfair comparison. The authors claim that existing models "fail to execute simple, direct instructions" due to a lack of "robust, fundamental navigation capabilities." This conclusion may be flawed because the observed failure is likely a result of OOD generalization issues since these models are trained exclusively on long, complex instructions (R2R/RxR distribution). They will naturally perform poorly when tested on a new distribution of short, isolated, primitive instructions (VLMB). The performance gap is a data shift problem, not necessarily a fundamental architectural flaw. So I would like to see the performance of some training-free methods that don't have this problem.
> >
> > Second, the new results confirm that the performance gain is primarily data-driven, not method-driven. The proposed Move-to-Anything model, when trained without the VLMB data, performs significantly worse than the baselines. The performance only becomes comparable after adding the 40K samples from VLMB. This confirms that the improvement stems from data augmentation rather than a superior architectural solution to general navigation.
> >
> > Finally, the initial review requested a broader comparison of existing approaches to properly support the claim that "existing models fail to execute these general but straightforward instructions effectively." The rebuttal only adds a discussion of UniGoal without further results, which is insufficient. I think more mainstream, high-performing VLN-CE baselines on the VLMB dataset should also be added.

---

> > > ### Author Response · Authors · 2025-11-28
> > > **Response to Reviewer XQBp (Training-free methods)**
> > >
> > > We thank the reviewer for raising this important point. We are aware of the existing work demonstrating that certain proprietary MLLMs possess **non-trivial high-level navigation decision-making capabilities** in some contexts [8-10]. Our intention is not to contradict these findings but to highlight a different, more fundamental issue.
> > >
> > > Following your suggestion, we evaluated GPT-4o and GPT-5-mini under a training-free setting. The results (see **Table 2**) confirm that both 7B models and these advanced commercial models perform poorly on our basic navigation task. The fact that GPT-5-mini underperforms even some 7B models is particularly indicative.
> > >
> > > - Table 2
> > > | Method | R-Nav | | | | | | | | VLMB | | | | | | | |
> > > |:---|:---:|:---:|:---:|:---:|:---:|:---:|:---:|:---:|:---:|:---:|:---:|:---:|:---:|:---:|:---:|:---:|
> > > | | **MP3D** | | | | **HM3D** | | | | **MP3D** | | | | **HM3D** | | | |
> > > | | SR↑ | SPL↑ | OS↑ | NE↓ | SR↑ | SPL↑ | OS↑ | NE↓ | SR↑ | SPL↑ | OS↑ | NE↓ | SR↑ | SPL↑ | OS↑ | NE↓ |
> > > | Vedio-LLaVA | 14.1 | 12.5 | 34.7 | 4.91 | 15.2 | 14.6 | 43.1 | 5.02 | 13.4 | 12.3 | 34.3 | 4.96 | 15.3 | 14.1 | 39.2 | 5.21 |
> > > | LLaVA-NEXT| 12.1 | 10.5 | 31.5 | 5.46 | 14.2 | 13.5 | 39.4 | 5.45 | 11.2 | 10.5 | 33.1 | 5.34 | 12.3 | 11.5 | 35.7 | 5.56 |
> > > | GPT-4o | 11.6 | 11.4 | 13.6 | 5.23 | 22.6 | 22.5 | 24.8 | 4.53 | 15.3 | 15.2 | 16.9 | 4.98 | 20.29 | 20.29 | 20.45 | 4.44 |
> > > | GPT-5-mini | 14.1 | 14.1 | 14.2 | 5.02 | 25.4 | 25.3 | 27.1 | 4.29 | 16.1 | 16.1 | 17.1 | 4.92 | 25.77 | 25.77 | 26.5 | 4.36 |
> > > | NaVid  | 46.7 | 42.6 | 59.8 | 4.74 | -- | -- | -- | -- | 43.8 | 39.0 | 60.1 | 4.89 | -- | -- | -- | -- |
> > > | StreamVLN  | 15.8 | 10.8 | 52.2 | 8.48 | 32.4 | 28.2 | 49.3 | 5.43 | 22.0 | 16.0 | 60.9 | 7.24 | 44.2 | 39.0 | 59.3 | 4.43 |
> > > | **Move-to** | **62.5** | **61.4** | **69.1** | **3.09** | **70.6** | **69.6** | **75.8** | **2.70** | **60.6** | **59.6** | **67.6** | **3.21** | **71.4** | **70.3** | **76.3** | **2.71** |
> > >
> > > This allows us to refine and strengthen our original claim. We posit that while advanced MLLMs may exhibit promising high-level planning abilities, they critically **lack low-level, foundational navigation competencies(e.g., precise action execution based on perceptual grounding)**. This distinction is crucial. It suggests that directly applying these models to long-horizon navigation tasks—which implicitly assume solid foundational skills—is an ill-posed approach. The model may plan a route but fail at the basic steps to execute it. Our work addresses this specific gap by first instilling these foundational capabilities.
> > >
> > > We believe this clarification significantly strengthens our paper's contribution. Thank you again for this insightful comment.
> > >
> > > **Reference**
> > >
> > > [8] Dang, Ronghao, et al. "Rynnec: Bringing mllms into embodied world." arXiv preprint arXiv:2508.14160 (2025).
> > >
> > > [9] Yang, Haolin, et al. "NavSpace: How Navigation Agents Follow Spatial Intelligence Instructions." arXiv preprint arXiv:2510.08173 (2025).
> > >
> > > [10] Qiao, Yanyuan, et al. "NavBench: Probing Multimodal Large Language Models for Embodied Navigation." arXiv preprint arXiv:2506.01031 (2025).

---

> ### Author Response · Authors · 2025-11-24
> **Response to Reviewer XQBp**
>
> Thank you for your thoughtful feedback! We appreciate your emphasis on data distribution and fairness as the core issue. Our work centers on **Navigation with Multi-Modal Large Language Models (MLLMs)**, addressing the limitation that current Video-LLM-based VLN models fail to execute general yet straightforward instructions effectively. A key contribution of our study is the introduction of an efficient **data construction pipeline** that enhances navigation capability in VLN models.
>
> First, we identify a fundamental limitation in the prevailing VLN training paradigm: existing approaches rely on long-horizon instruction (R2R/RxR) training to impart navigation ability. However, our analysis shows that these models fail to learn basic navigation primitives. Visualization experiments (Appendix B.2, Fig. 8) reveal that even successful cases under current paradigms do not follow intermediate sub-goals sequentially; instead, they overfit to the final goal location. This indicates a **systemic flaw in the VLN training paradigm**. We are the first to highlight this issue and propose a viable solution. Table 2 further demonstrates that training-free models such as LLaVA-Vid and LLaVA-Next perform poorly on R-Nav and VLMB tasks, with navigation success rates ranging only from **11.2% to 15.3%**, reinforcing the need to rethink the current paradigm.
>
> Second, regarding the role of data, we agree that VLN research has been predominantly data-driven, and high-quality data remains the most effective means of improvement. From NaVid [2] to recent models like NaVILA [7] and StreamVLN [3], the focus has been on leveraging large-scale external VLN-CE data to boost performance. For example, NaVid incorporates 953K additional samples at a cost of 672 A100 GPU hours, while StreamVLN uses 10,033K samples requiring 1,500 A100 GPU hours. In contrast, our approach achieves competitive gains with only 40K samples (76 A100 GPU hours), underscoring the efficiency of our pipeline. If the improvement were purely data-driven, our model—**with 1/25th the data of competitors**—should perform significantly worse. The fact that it competes with models trained on millions of samples demonstrates the effectiveness of both our data and method.
>
> Finally, our discussion of UniGoal [1] illustrates that other navigation models lack general language understanding capabilities. Importantly, our statement that existing models fail to execute general but straightforward instructions refers specifically to **navigation models trained under MLLM frameworks**. Among VLN-CE baselines, NaVid (2024 SOTA) and StreamVLN (Sept 2025 SOTA) already outperform other VLNCE-trained models significantly. If broader comparisons are deemed necessary, we are prepared to include additional baselines in future revisions.
>
> **Reference**
>
> [7] Cheng, An-Chieh, et al. "Navila: Legged robot vision-language-action model for navigation." Proceedings of Robotics: Science and Systems. 2025.

---

### Official Review · Reviewer_Nwdw · 2025-10-30

**Soundness:** 2
**Presentation:** 2
**Contribution:** 2
**Rating:** 4
**Confidence:** 4

**Summary:**

This paper focuses on the field of Vision-Language Navigation (VLN) and points out that although current end-to-end VLN models based on Multi-modal Large Language Models (MLLMs) perform reasonably well on long-horizon instruction tasks, they have significant shortcomings in basic navigation primitives (such as moving and changing regions). To address this issue, the paper proposes a primitive-based learning paradigm, which first learns core skills and then combines them into long-horizon navigation behaviors. It constructs the first controllable benchmark dataset, Vision-Language-Move-Base (VLMB), centered on the "move-to" primitive, covering 206 scenes and 873 object instances. Based on this dataset, the paper develops the Move-to-Anything model with a memory mechanism that balances historical context and current observations. Experiments show that this method achieves a success rate of 60.6% in the MP3D scene and 71.4% in the HM3D scene, significantly outperforming the 43.8% success rate of existing models, and demonstrating stronger compositional generalization ability.

**Strengths:**

1. It identifies the performance shortcomings of end-to-end VLN models in basic navigation primitives, breaking the inherent perception that "good performance on long-horizon tasks implies solid basic capabilities".

2.  A unified data pipeline is designed to build the VLMB dataset. Focusing on the "move-to" primitive, it integrates multi-scene resources, enhances spatial-semantic information, and ensures data quality through interactive verification. It fills the gap of existing datasets in training basic navigation skills and provides reliable data support for subsequent related research.

3. The proposed hierarchical memory mechanism effectively balances recent detail fidelity and global context by decomposing historical observations into short-term and long-term components. At the same time, temporal and segment embeddings are introduced to improve the stability of long-horizon reasoning.

4. From problem formulation, literature review of related work, to dataset construction, model design, and experimental verification, each part is coherently connected, and the argumentation progresses step by step. This allows readers to clearly understand the research ideas and technical paths, ensuring high readability.

**Weaknesses:**

1. Although the paper verifies the model's performance in MP3D and HM3D scenes, it does not deeply explore the model's performance in more complex and diverse real physical environments (such as extreme weather and scenes with dense dynamic obstacles). It is recommended to supplement test results in such scenes, provide failure case analyses and feature visualization, to help readers clearly grasp the applicable scope and limitations of the method.

2. Existing ablation experiments verify the effectiveness of dataset cross-validation, Hierarchical Memory (HM), and Temporal & Segment Embeddings (TSE). However, there is a lack of sensitivity analysis on parameters such as the short-term window size (W) and the number of long-term memory slots (M) in HM, as well as disassembly experiments on the respective contributions of temporal embeddings and segment embeddings in TSE. It is recommended to supplement such experiments to clarify the optimal settings of each parameter and the independent role of each component.

3. The paper does not mention the comparison of the Move-to-Anything model with existing SOTA VLN models (such as NaVid and StreamVLN) in terms of parameter count, training duration, and inference speed. It is recommended to supplement quantitative analysis of model complexity and efficiency.

**Questions:**

See weaknesses 2.

---

> ### Author Response · Authors · 2025-11-20
> **Response to Reviewer Nwdw**
>
> We thank the reviewer for the constructive feedback. Below are our responses to the major concerns raised. Any further discussion will be appreciated. Besides, we have updated the PDF of our paper to address your concerns.The corresponding modified sections in the manuscript (see PDF in Supplementary Material) have been marked in Blue.
>
> >**Weakness1: Generalization to dynamic obstacles and extreme weather**
>
> - Thank you for raising this important point. We would like to clarify that Video-LLM-based VLN models (e.g., NaVid [1], streamVLN [2]) primarily target semantically rich indoor environments, whereas dynamic obstacles and extreme weather are more characteristic of outdoor settings with sparse semantics. The primary goal of VLN research is to enable agents to interpret diverse human instructions and execute navigation actions accurately in **environments with dense semantic cues**.
>
> - The core motivation of our work is to address a concerning trend in the VLN community: existing models increasingly prioritize visual-text alignment in long-horizon tasks while overlooking fundamental navigation skills. To refocus on navigation competence, we designed a comprehensive automated data generation pipeline and trained a model to validate its effectiveness.
> Importantly, a major part of our motivation is to equip MLLMs with true all-weather and all-terrain navigation capabilities. We fully acknowledge the significance of real-world challenges and plan to extend our framework to more diverse and complex conditions in future work.
>
> >**Weakness2: Ablation studies on Hierarchical Memory**
>
> - We have supplemented ablation studies on the short-term window size (W) and long-term memory slots (M) in **Section 5.5 and Table 6**. Results show that increasing both W and M improves navigation performance up to an optimal configuration (W=8, M=1568), while the mean+MLP aggregator achieves a favorable balance between **hierarchical memory preservation and computational efficiency**. The reason we did not conduct an isolated ablation for HM (i.e., TSE without HM) is that the Temporal \& Segment Embeddings module is designed to depend on HM, as it incorporates **temporal and semantic embeddings based on HM's token selection**. The two components are inherently coupled in our design.
>  | W  | M    | Aggregator Type | MP3D | | | | | HM3D | | | |
> |----|------|-----------------|------|-|-|-|-|------|-|-|-|
> |    |      |                 | SR↑ | SPL↑ | OS↑ | NE↓ | | SR↑ | SPL↑ | OS↑ | NE↓ |
> |----|------|-----------------|-----|------|-----|-----|-|-----|------|-----|-----|
> | 4  | 784  | Mean+MLP        | 53.2 | 51.8 | 60.3 | 3.78 | | 64.7 | 63.1 | 69.2 | 3.12 |
> | 8  | 784  | Mean+MLP        | 57.1 | 55.7 | 64.4 | 3.45 | | 68.9 | 67.5 | 73.4 | 2.89 |
> | 8  | 1,568| Mean+MLP        | **60.6** | **59.6** | **67.6** | **3.21** | | **71.4** | **70.3** | **76.3** | **2.71** |
> | 16 | 1,568| Mean+MLP        | 58.8 | 57.9 | 65.9 | 3.38 | | 70.6 | 69.4 | 75.1 | 2.74 |
> | 8  | 1,568| N/A             | 57.7 | 56.5 | 64.2 | 3.40 | | 68.9 | 67.6 | 73.8 | 3.01 |
>
> >**Weakness3: Quantitative analysis of model complexity and efficiency**
>
> - Thank you for raising this point. To ensure a fair comparison, our model was trained with the same backbone (LLaVA-Vid [3]) and identical parameter settings as NaVid and StreamVLN. We provide additional comparisons in **Appendix B.2  and Table 8**, including dataset size, training time, and inference speed. Results show that our approach requires significantly **less training data, reducing training cost, and achieves faster inference** thanks to the streamlined instructions in VLMB.
> | Method | Training dataset | Dataset size | Training time (GPU hours) | R2R Inference time (s) | VLMB Inference time (s) |
> |--------|------------------|--------------|---------------------------|------------------------|-------------------------|
> | NaVid [1] | R2R+RxR+VLNdata | 953K | 672 | 7.9638 | / |
> | StreamVLN [2] | R2R+RxR+EnvDrop | 10033K | 1500 | 8.3288 | / |
> | Move-to-Anything (ours) | VLMB | 40K | 76 | / | 3.0852 |
>
> We sincerely appreciate the reviewer's insightful comments, which have been instrumental in enhancing the quality of our manuscript.
>
> **Reference**
>
> [1] Zhang, Jiazhao, et al. "Navid: Video-based vlm plans the next step for vision-and-language navigation." Proceedings of Robotics: Science and Systems. 2024.
>
> [2] Wei, Meng, et al. "Streamvln: Streaming vision-and-language navigation via slowfast context modeling." arXiv preprint arXiv:2507.05240 (2025).
>
> [3] Zhang, Yuanhan, et al. "Video instruction tuning with synthetic data." arXiv preprint arXiv:2410.02713 (2024).

---

> ### Author Response · Authors · 2025-11-27
> **Response to Reviewer Nwdw**
>
> Dear Reviewer,
>
> As the rebuttal discussion phase is coming to an end, we would like to confirm whether we have correctly addressed the concerns you raised. If you have any additional questions or feedback, please let us know at your earliest convenience.
> Your input is extremely valuable to us, and we are committed to resolving any issues and improving our work based on your suggestions.
>
> Wishing you a Happy Thanksgiving and a wonderful day!
>
> Best regards

---

### Author Response · Authors · 2025-11-20
**Global Response**

**Dear SAC, AC and Reviewers,**

We thank all reviewers and the AC for their constructive feedback. We are encouraged by the positive comments about the strengths of our paper in the review process:

-  The highlighted phenomenon is **interesting and significant in current VLN-oriented VLA methods**, pushing the direction of VLN research toward more open-ended and human-like reasoning rather than narrow, dataset-dependent training. (Review Nwdw, XQBp, NPKL)

- The data could help the community **better evaluate methods in this field**. It fills the gap of existing datasets in training basic navigation skills and provides reliable data support for subsequent related research. (Review Nwdw, NPKL)

- The Move-to-Anything framework is **largely technically sound and demonstrates better performance** than existing methods. The results validate that the method is better at handling flexible, goal-driven instructions. (Review XQBp, NPKL)

- The paper is **clearly presented and visualized**. Each part is coherently connected, and the argumentation progresses step by step. This allows readers to clearly understand the research ideas and technical paths, ensuring high readability. (Review Nwdw, XQBp)

Further, we summarize our response to reviews to clarify some misconceptions and strengthen our claims:

- **[Clarification of Motivation]**

    We pinpoint a fundamental flaw in the mainstream end-to-end VLN training paradigm: its reliance on long-instruction sequences leads models to **overfit to visual-textual cues at the expense of learning basic navigation skills**. This is evidenced by our visualization experiments (**Appendix B.2, Fig. 8**), which show that even "successful" case fail to follow instructions stepwise. To address this, our work refocuses on strengthening foundational navigation abilities by constructing new datasets and benchmarks to promote general-purpose navigation model development.

- **[Comparison Fairness]**

    **Fairness in Dataset Evaluation**:​ We have supplemented a detailed analysis of the instruction composition in VLN-CE (**Appendix B.1**). The analysis shows that the R-Nav-MP3D (e.g., "move to the desk") essentially forms an instructional subset of VLN-CE, and such instructions are already included in existing model training sets.

    **Fairness in Method Comparison**:​ To further validate the contribution of the VLMB dataset to navigation capability, we conducted joint training experiments using VLMB together with VLN-CE. Experimental results (**Table 4**) show that after incorporating a small amount of VLMB data (40k vs. 10053k), our method achieves improved performance on the R2R benchmark (+4.4). Following suggestions, we also evaluated GPT-4o and GPT-5-mini in a training-free setting. The results (**Table 2**) show that both 7B models and advanced commercial models perform poorly on the basic navigation task.

- **[Extended Experimental Results]**

    Add an ablation study on the HM parameters to demonstrate the rationality of the parameter selection (**Table 6**).

    Add a quantitative analysis of the instruction composition to support the instruction decomposition (**Table 7**).

    Add the computational costs of training and inference for analyzing model complexity and effectiveness. (**Table 8**).

In summary, this work is the first to pinpoint **a critical shortfall in generating basic navigation actions in end-to-end MLLM-based VLN models**. The Move-to-Anything model using the goal-oriented VLMB dataset demonstrates the efficacy of the **primitive-based step-by-step learning paradigm in developing robust and generalizable navigation agents**.

Best regards,

The Authors

---

### Meta-Review · Area_Chair_1nsi · 2026-01-02

**Summary:**

This paper presents a approach to Vision-Language Navigation by identifying a shortcoming in current end-to-end VLN models: while they perform reasonably well on long-horizon navigation tasks, they struggle with basic navigation primitives like "move-to" instructions. The authors propose a primitive-based learning paradigm, introduce the VLMB dataset focused on the "move-to" primitive, and develop the Move-to-Anything model with hierarchical memory mechanisms.

Reviewers raised several key concerns that informed the decision-making process. Reviewer Nwdw questioned the model's generalization to complex real-world environments, requested sensitivity analysis of hierarchical memory parameters, and asked for quantitative analysis of model complexity. Reviewer XQBp challenged the motivation and fairness of comparisons, arguing that performance gaps might stem from data distribution differences rather than fundamental architectural flaws, and questioned the novelty of the approach. Reviewer NPKL sought deeper analysis of instruction length impacts, clarification of memory mechanisms, and validation on standard VLN-CE benchmarks.

**Reviewer Concerns:**

The authors provided responses that addressed many concerns. However, Reviewer XQBp's core concern about fairness and data distribution remains partially outstanding. While the authors added training-free evaluations of GPT-4o and GPT-5-mini and provided analysis showing R-Nav instructions are subsets of VLN-CE, the reviewer maintained that the performance gap might still reflect out-of-distribution generalization issues rather than fundamental skill deficiencies. The authors' argument that their approach achieves competitive results with only 1/25th of competitors' data is compelling but doesn't fully resolve the distribution concern.

**Reviewer Scores:**

Based on the rebuttal, the AC think the reviewers would have adjusted their scores as follows:

Reviewer Nwdw might maintain 4 or increase to 5. The authors addressed all three weaknesses: added sensitivity analysis, computational cost comparisons, and clarified generalization scope.

Reviewer XQBp might maintain 4. While the authors provided substantial additional analysis and experiments, the core philosophical disagreement about whether the observed phenomenon represents a fundamental skill gap versus a data distribution issue remains. The addition of training-free model evaluations strengthens the authors' position, but the reviewer might still view this as an incremental contribution.

Reviewer NPKL maintains 6. The reviewer explicitly stated "my concerns have been addressed" and maintained the original score.

Therefore, the overall score tends to be a rejection.

---

### Decision · Program_Chairs · 2026-01-26

Reject